# AI-for-Science Low-code Platform with Bayesian Adversarial Multi-Agent Framework

**Zihang Zeng**[1,2#]**, Jiaquan Zhang**[1,2#]**, Pengze Li**[1]**, Yuan Qi**[1,3]**, Xi Chen**[1,3†]
[1]Artificial Intelligence Innovation and Incubation Institute, Fudan University
[2]Shanghai Innovation Institute
[3]Shanghai Academy of AI for Science
`x_chen@fudan.edu.cn`

## ABSTRACT

Multi-agent systems leveraging Large Language Models (LLMs) show immense potential for solving complex scientific problems. However, their reliability is undermined by the probabilistic nature of LLMs, which can produce hallucinations in both generated code and its corresponding test cases. In a multi-agent architecture, these errors can propagate and compound, leading to flawed final outputs. To overcome these core limitations, we introduce a novel Bayesian Adversarial Multi-agent Framework for AI for Science (AI4S). Delivered as a Low-code Platform (LCP), our framework enhances the coding capability for scientific tasks across a wide range of base models, from 1.7B open-source LLMs to up-to-date commercial ones. Our framework employs three agents in a recursive loop that adversarially co-optimizes the generated solutions, the test cases used for evaluation, and the prompts driving generation. This process is governed by a non-LLM-based Bayesian updating rule, which systematically reduces evaluation uncertainty and mitigates the system's dependence on any single LLM's reliability. Furthermore, the LCP empowers domain experts by translating high-level natural language prompts into executable, domain-specific requirements, eliminating the need for intricate prompt engineering. Extensive experiments confirm that our framework generates robust solutions while effectively minimizing error propagation. On a complex, cross-disciplinary Earth Science benchmark, our platform demonstrates superior reliability and outperforms state-of-the-art models, where a 32B open-source model can beat the performance of a 235B model in the ScienceCode benchmark with our framework.

## 1 INTRODUCTION

Large Language Models (LLMs) are transforming AI for Science (AI4S) research paradigm by automating complex scientific code generation for simulations, data analysis, and related science tasks (Chowdhery et al., 2023; Nijkamp et al., 2022). While models such as Codex, AlphaCode, and CodeLlama effectively lower technical barriers for researchers (Chen et al., 2021; Li et al., 2022; Roziere et al., 2023), several challenges hinder their reliable application in AI4S research. These include: (1) potentially unclear prompt descriptions from domain scientists without computer science backgrounds, (2) complex execution pipelines for scientific tasks, and (3) the need to maintain adherence to physical laws and domain-specific constraints. Standard prompting and self-refinement techniques (Wei et al., 2022; Chen et al., 2023; Olausson et al., 2023) are often inadequate for handling the subtle error patterns in complex scientific workflows.

Crucially, we lack strong empirical evidence to fully trust LLMs' capabilities in deep understanding and complex reasoning, particularly for professional scientific research tasks (Ridnik et al., 2024). Their decision-making processes remain opaque, and while their outputs often appear plausible, they may contain subtle inaccuracies or conceptual misunderstandings. These limitations fundamentally

---

[#]Equal contribution. [†]Corresponding author.

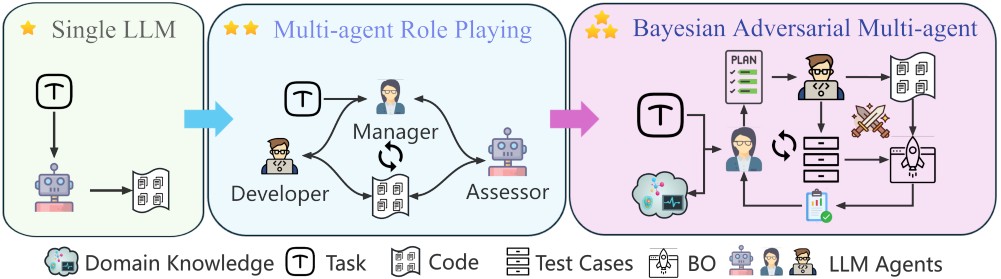

Figure 1: Comparison between three code generation paradigms: Single LLM generator, multi-agent role playing and the proposed Bayesian adversarial multi-agent framework.

constrain the performance ceiling of LLM-based coding platforms, as their capabilities are inherently bounded by the underlying LLM's intelligence level. Such inherent uncertainty demands the development of frameworks that operate without requiring absolute confidence in the LLM's intelligence level.

As illustrated in Figure 1, recent advances in LLM-based multi-agent systems attempt to address these limitations through distributed reasoning and specialized agent roles, where different LLM agents focus on specific sub-tasks while coordinating through structured communication protocols and/or a master agent (the green ellipse). However, while such multi-agent architectures (Hong et al., 2023; Wu et al., 2023) can effectively distribute computational complexity and domain expertise across components of the underlying domain task, they introduce new challenges in error propagation and validation. The system's overall reliability becomes constrained by its weakest agent, as flawed intermediate outputs from one of the agents can be uncritically accepted by downstream agents (Huang et al., 2024), potentially amplifying rather than mitigating the above limitations of individual LLMs. Furthermore, evaluating the code of domain-specific tasks is often difficult. Standard unit tests may miss critical scientific constraints, theoretical foundations, or domain-specific limitations. This evaluation gap stems from three key issues: (1) scientific correctness often requires deeper domain knowledge than standard unit tests can verify; (2) comprehensive evaluation metrics may be prohibitively expensive or fundamentally intractable to define; and (3) LLM-generated tests may inherit the same reliability issue as the code they aim to validate (Zhou et al., 2023). Given these, one must pay equal attention to both LLM-generated code and the test cases used to assess it.

This fundamental insight motivates our core design philosophy: an adversarial co-evolution framework where test case generation and code improvement mutually refine each other through competitive optimization, replacing traditional static verification approaches. The proposed framework structures agent interactions and evolves prompt distributions using Bayes' Theorem, reducing dependence on the base LLM's inherent capabilities. The framework comprises three specialized agents: a Task Manager (TM) serving as Challenger, a Solution Generator (SG) as Solver, and an Evaluator for comprehensive assessment. Unlike conventional multi-agent code generation systems that depend entirely on LLM-based evaluation and decision-making(Qian et al., 2023; Hong et al., 2023), our approach introduces an adversarial dynamic between TM and SG. The TM actively constructs and refines test cases to probe the SG's current limitations, while the SG iteratively improves its code generation based on Evaluator feedback to meet these evolving challenges. As shown in Figure1 orange ellipse, by continuously probing and validating solutions against dynamically refined test cases, our framework not only overcomes these evaluation barriers but also progressively converges on solutions that satisfy both explicit requirements from domain experts and implicit domain constraints from the specific domain or application scenarios.

The proposed framework also enhances Human-AI collaboration in AI4S community(Yamada et al., 2025; Zheng et al., 2025; Babaei Giglou et al., 2024). Outside the Machine Learning community, we cannot expect an average scientist to be aware of, let alone skilled in, the extensive list of prompt engineering techniques. A typical domain researcher's prompt might be vague, assume implicit domain knowledge, or use specialized terminology and abbreviations that an LLM, especially a smaller one, may not fully grasp. These domain gaps may lead to misinterpretations, suboptimal outputs, or complete system failures. To bridge this gap, our framework incorporates a specialized scheme

within TM agent that actively structures raw user requests, resolves ambiguities through interactive clarification, and transforms potentially vague prompts into precise task plans and scientifically valid initial test cases. It maintains accessibility for non-technical domain experts while fully leveraging their domain expertise without requiring any computer science or professional prompt engineering skills. The main contributions of this work are threefold:

- **A Novel AI4S Low-Code Platform with Bayesian Adversarial Framework:** We introduce a multi-agent framework that employs a Bayesian recursive co-updating strategy to iteratively refine generated code and test cases using a non-LLM-based adversarial score. This method significantly enhances scientific coding performance across a spectrum of base models (from open-source to commercial) and allows smaller LLMs to achieve results competitive with larger counterparts.

- **Bayesian Optimization for code performance estimation:** We proposed a Bayesian Optimization method to estimate the performance of a given code based on its structure similarity with the tested codes, which enables the framework to handle and evaluate complicated code.

- **Domain Knowledge Refinement for scientific tasks:** The LCP facilitates scientific exploration for non-coding professionals by enabling the generated code to better reflect domain knowledge and constraints through iteratively refining, adding and updating domain knowledge in the specially structured prompt. Our Earth Science case study exemplifies this, where the generated machine learning model not only produced superior predictions but also demonstrated minimal deviation from established ocean dynamics, ensuring scientific consistency.

In the rest of this paper, we introduce the main methodology and models in Section 2, followed by experimental setup and numerical results in Section 3. We conclude the work and discuss its future work and limitations in Section 4.

## 2 METHOD

### 2.1 OVERVIEW

We propose a Bayesian adversarial multi-agent framework designed for AI4S tasks, incorporating subjective prior knowledge and addressing complex task abilities. The framework comprises three core component agents: a Task Manager (TM), a Solution Generator (SG), and an Evaluator(Eval). Within this structure, code generation becomes a dynamic interaction, primarily between the Task Manager (acting as a Challenger) and the SG agent (acting as a Solver), with the Evaluator providing the performance metrics that guide learning and adaptation. The game concludes when the SG agent produces code that successfully passes all defined validation tests.

The process initiates prior knowledge $\mathcal{P}$ with a task description (provided by a scientist user) and relevant subject materials (e.g., prior domain knowledge, including reference code samples). $\mathcal{P}$ is the initial input to the TM agent, which develops a structured plan of the scientific task, decomposing the main task into an ordered set of Sub-tasks. This plan is iteratively refined based on users' feedback $\mathcal{F}$ and refinements until user's approval and denoted as *Plans*, as indicated by Loop 1 of Figure 2. Subsequently, the TM agent generates an initial set of test cases (*Test Case$_0$*) corresponding to these sub-tasks and other criteria derived from prior knowledge. These initial test cases, along with user-provided reference code base are serving as initial sample codes (*Sample Code$_0$*). *Sample Code$_0$* is followed by the user approved plan (*Plans*) to form the initial prompt:

$$Prompt_0 := Plans \oplus Test\ Case_0 \oplus Sample\ Code_0, \tag{1}$$

where $\oplus$ is the direct concatenation operator. Both the test cases and the sample codes can be independently updated in subsequent iterations. This update mechanism, guided by Bayesian principles, leverages the performance of candidate codes generated previously and the effectiveness of past test cases. The objective is to iteratively refine the prompt to guide the SG agent towards producing a solution that meets all test criteria and user requirements. The core Bayesian update rule for selecting a specific test case $i$ and sample code $j$ for the prompt at iteration $t + 1$ is: $p(Prompt_{ij}^{t+1}|S_3^t) \propto p(S_3^t|Prompt_{ij}^t)p(Prompt_{ij}^t)$. This iterative refinement continues until the SG

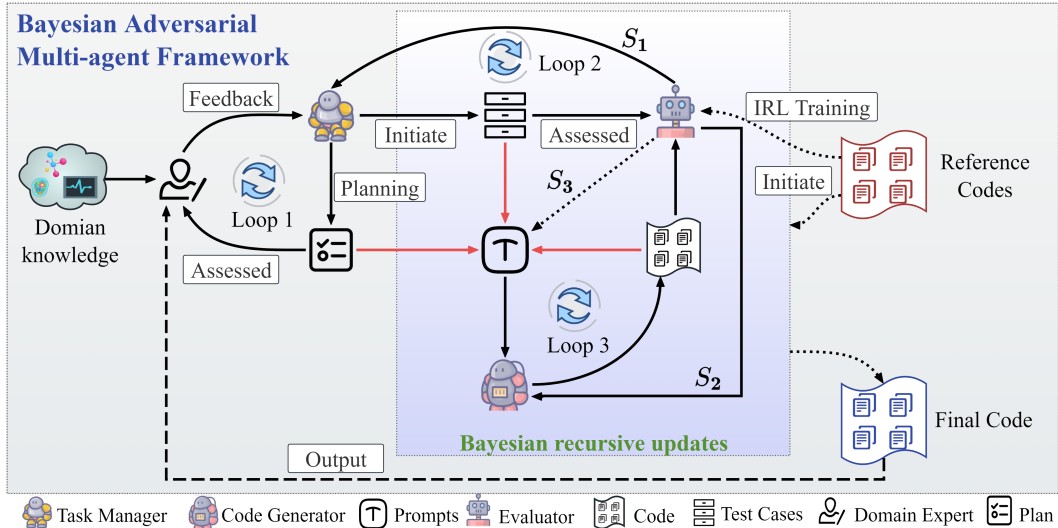

Figure 2: Overview of the Bayesian adversarial multi-agent framework. The three red arrows indicate fusion of the user-approved plan, test cases, and codes into prompts, the distribution of which is recursively updated under the Bayesian framework. $S_1$, $S_2$ and $S_3$ are the scores computed in equation 2, equation 3, and equation 4. Loop 1-3 indicate three iterative updating processes for plan, test cases, and codes, respectively. The dashed arrows indicate latent relationships (e.g., $S_3$ likelihood score) or steps conducted before or after the main algorithm execution.

agent achieves satisfactory success on the test cases. The pseudo-code of the proposed method is described in Algorithm 1.

---

**Algorithm 1** Bayesian Adversarial Multi-Agent Framework

1: **Input:** Task description and domain knowledge $\mathcal{P}$, reference code base, max iterations $T_{max}$
2: // Planning till User's approval
3: Generate $Plans = \text{TM}(\mathcal{P})$
4: **while** NOT User Approval **do**
5:     $Plans \leftarrow \text{TM}(\mathcal{P}, \mathcal{F})$ iteratively updates the plan given user feedback.

6: // Make initial prompts
7: Generate $Test\ Case_0 := (\{\text{Sanity Checks}\}, \{\text{Sub-tasks}\}) \leftarrow \text{TM}(Plans)$
8: Form $Sample\ Code_0 \leftarrow \{Test\ Case_0, Reference\ Code\}$
9: Generate $Prompt_0$ according to equation 1
10: // Code generation and evaluation
11: $t \leftarrow 0$
12: Initialize test case scores $\forall i \in \{1, \ldots, M\}, S_1(i)^0 \leftarrow 1$
13: **while** Not $\exists$ Code : pass all checks $\land t < T_{max}$ **do**:
14:     $t \leftarrow t + 1$
15:     Generate code batch $\{\mathbf{C}_j\}_{j=1}^N \leftarrow SG(Prompt_{t-1})$
        // Score Calculation & Updates
16:     Compute Code Scores $S_2(j)^t$ using $S_1^{t-1}$ according to Eq. equation 3
17:     Update Test Case Scores $S_1(i)^t$ using $S_2(j)^t$ according to Eq. equation 2
18:     Compute Prompt Score $S_3^t$ using $S_1^t, S_2^t$ according to Eq. equation 4
19:     $C_{best} = \arg\max_j S_2(j)^t$
        // Bayesian updates for next iteration
20:     $Test\ Case_t \leftarrow \text{TM}(Test\ Case_{t-1}, S_1^t)$         ▷ Refine test cases based on $S_1^t$
21:     $Prompt_t \sim p(Prompt_t | S_3^t)$         ▷ Update prompt based on $S_3^t$
    **return** $C_{best}$

---

## 2.2 PLANNING AND INITIAL CODE GENERATION

As briefly described in the above overview, the TM agent engages in a comprehensive planning phase. In particular, this involves: 1. Decomposing the primary task into sub-tasks. 2. Posing Sanity Checks for data structure and range. 3. Reasoning about the logical workflow and dependencies between sub-tasks. 4. Formulating strategic advice for the generation of effective test cases. This detailed plan is presented to the user in natural language for review and potential refinement. This interactive feedback loop continues until the user approves the plan. Once the plan is finalized, the TM agent generates an initial set of test cases. These test cases are designed to cover the specified sub-tasks and incorporate domain-specific prior knowledge, such as sanity checks (e.g., for expected data ranges) and out-of-range detection.

Following the planning phase, the system constructs the initial prompt *Prompt*$_0$ as defined in equation 1. The code generation agent then uses this prompt to produce N candidate solutions (codes). For each candidate, the system automatically generates comprehensive documentation containing: algorithm explanations, execution instructions, and detailed specifications for all functions, variables, and parameters.

## 2.3 A PRIORI ESTIMATION WITH BAYESIAN OPTIMIZATION

We noticed that executing all the generated codes for testing can be computationally expensive. To address this issue, as well as to leverage both the accuracy of evaluation and the range of exploration in the solution space, a Bayesian Optimization method is employed to estimate the performance score relates to the structural difference from all the tested codes (as in Loop 3 of Figure 2).

As an initiation, all the generated code in the first iteration $\{Code_i\}_{1 \leq i \leq N}$ get tested against the initial test cases, where all test cases start from the same initial difficulty score ($S_1(i)^0 = 1$). We store the test results as a score vector $S_2$ (will be explained in details in equation 3). We then embed each $Code_i$ to a vector $\mathbf{x}_i$ through a structural embedding that captures features from its Abstract Syntax Tree (AST) and code embedding vectors. We then use a Bayesian optimization process to predict the code's performance based on its structural similarity with the tested code, which is detailed explained in the appendix. This Bayesian Optimization approach allows the system to intelligently explore the vast solution space, prioritizing the most promising candidates for expensive testing and efficiently converging towards a high-quality solution. It supports the evolution of the distribution of prompt, thus the co-evolution of codes and test cases.

## 2.4 EVALUATION AND FEEDBACK

The Evaluator agent is responsible for assessing the candidate codes, the effectiveness of the test cases, and the overall quality of the prompts used in each iteration.

**Test Case Score** ($S_1$): This score quantifies the "True Hardness" of the $j$th test case *Test Case*$_j$, representing its capacity to be challenging yet ultimately solvable. An effective test case should successfully discriminate between code solutions of varying quality (as in loop 3 of Figure 2).

$$S_1(i)^{t+1} = (1 - \alpha) \cdot S_1(i)^t + \alpha \cdot \left( \frac{\sum\limits_{j' \text{ s.t. pass}} S_2(j')}{|\{\text{Code}_{j'}\}|} - \frac{\sum\limits_{j^\dagger \text{ s.t. fail}} S_2(j^\dagger)}{|\{\text{Code}_{j^\dagger}\}|} \right) \tag{2}$$

where $S_1(i)^t$ is set as 1 by default for $t = 0$, and $\alpha$ is a hyperparameter to control the momentum of updating, which in experiments we set $\alpha = 0.8$.

**Code Score** ($S_2$): Each generated code $C_j : 1 \leq j \leq N$ receives a composite score based on several factors:

$$S_2(j)^t = \frac{\sum_i \mathbb{I}(\mathbf{C_j} \text{ passes } T_i) \cdot S_1(i)^{t-1}}{\sum_i S_1(i)^{t-1}} \tag{3}$$

**Prompt Score** ($S_3$)**:** The overall score for a prompt used in an iteration is a function of the performance of the codes and test cases generated by this prompt:

$$S_3{}^t = \frac{1}{M} \sum_{j=1}^{M} S_1(i)^t + \frac{1}{N} \sum_{i=1}^{N} S_2(j)^t \tag{4}$$

If multiple prompt configurations are tested within a single logical iteration, the iteration's representative prompt score might be the highest achieved. For the Bayesian update, we are interested in the score of a specific prompt configuration $\{Prompt_{ij}^t\}_{ij}$ is then denoted $\{(S_3)_{ij}^t\}_{ij}$.

## 2.5 ITERATIVE REFINEMENT: ADVERSARIAL DYNAMICS AND BAYESIAN PROMPT UPDATES

The core of the framework's learning capability lies in its iterative refinement loop, characterized by an adversarial dynamic between the TM agent (Challenger) and the SG agent (Solver), and guided by Bayesian updates for prompt composition.

**Adversarial interaction:** The TM agent's role evolves to that of a Challenger. Based on the 'True hardness', which is measured by $S_1$, the TM adapts its weights for future evaluation and selects test cases for subsequent prompts. It aims to create test suites that are optimally challenging for the SG's current learned capabilities—difficult enough to drive further learning and expose weaknesses, yet generally solvable to provide a positive learning signal. The SG agent, as the Solver, implicitly adapts by producing code in response to these evolving challenges. Its success or failure provides the feedback signal that shapes the TM's subsequent challenging strategy.

**Bayesian prompt updates:** The selection of which specific test cases (indexed by $i$) and sample codes (indexed by $j$) to include in the prompt for the next iteration ($Prompt_{t+1}^{ij}$) is governed by a Bayesian update rule(as indicated by the combination of Loop 2 and 3 in 2):

$$p(Prompt_{ij}^{t+1}|S_3^t) \propto p(S_3^t|Prompt_{ij}^t)p(Prompt_{ij}^t) \tag{5}$$

Here:

- $p(Prompt_{ij}^t)$ is the prior probability of selecting the pair (*Test Case$_i$*, *Sample Code$_j$*) for the prompt. This prior can be uniform initially and can adapt over time based on the historical effectiveness of these components.

- $p(S_3^t|Prompt_{ij}^t)$ is the likelihood of observing the score $S_3^t$ given that the prompt was formed using *Test Case$_i$* and *Sample Code$_j$*. This term captures how well this specific combination performed. A potential formulation to ensure non-negativity and reflect that better-than-expected performance is more likely could be:

$$p(S_3^t|Prompt_{ij}^t) \propto \exp\left(\mathbb{E}[S_3{}^{t-1}\mathbf{1}(i,j)]\right) \tag{6}$$

where $\mathbb{E}[S_3\mathbf{1}(i,j)]$ is the expected score for the generated code with $Test_i, Code_j$ in the prompt based on past performance or a baseline. This implies that a prompt configuration performing significantly better than its historical average for that pair (*Test Case$_i$*, *Sample Code$_j$*) will have a higher likelihood.

The underlying intuition is to identify and prioritize "teacher-subject" pairs-specific combinations of sample code and test cases that consistently yield high-scoring prompts. This approach effectively learns which forms of guidance produce optimal results for different types of coding challenge.

**Sample code pool management:** The pool of available sample codes (*Sample Code*) is not static, but recursively updated as illustrated in Loop 3 of Figure 2. Initially, it contains user-provided reference codes. As the SG agent generates new codes $C_g^{(t)}$, those that achieve high $S_2$ can be added to the *Sample Code* pool. The selection of a *Sample Code$_j$* for a prompt can then be influenced not only by its initial status (as a reference) but also by an evolving measure of its "guidance quality," learned from its impact on past prompt scores when it was included.

**Final results:** Within each round, if there exists a code that can pass all the test cases, the System will output it as the final result to the user. Otherwise, the System will keep using the Bayesian Adversarial method recursively till generation of a satisfying code or reaching the maximum round of iterations, which is chosen by the user in the beginning and by default set to 3 by experience(See Section 3).

## 3 EXPERIMENTS

### 3.1 EXPERIMENTAL SETUP

**Benchmarks** To ensure a thorough evaluation, we utilize a diverse set of benchmarks. For general code generation, we use `HumanEval`, `HumanEval-ET`, `MBPP`, `MBPP-ET`(Austin et al., 2021; Dong et al., 2025; Hendrycks et al., 2021), and the more challenging `APPS`(Hendrycks et al., 2021) benchmark. For AI for Science tasks, we use the domain-specific `SciCode`(Tian et al., 2024) and `ScienceAgentBench`(Chen et al., 2024) benchmarks.

**Base Models** Our framework is designed to be model-agnostic. To demonstrate this, we integrate several backbone large language models (LLMs), including the `Qwen3` series (ranging from 1.7B to 235B)(Yang et al., 2025), which has versatile sizes, strong reasoning, it can demonstrate if our framework is also effective on the latest models. Beside, we also choose `Deepseek-v3`(Liu et al., 2024), `Deepseek-R1`(Guo et al., 2025), `Claude-sonnet-4`(Anthropic, 2025), `GPT-3.5-turbo`(Brown et al., 2020), and `GPT-4o`(Hurst et al., 2024). This allows us to assess the performance gains attributable to our framework across a spectrum of model capabilities.

**Compared Methods** We compare our framework against several state-of-the-art baselines. These include foundational strategies like `Few-Shot` prompting and `Chain-of-Thought (CoT)`, as well as other prominent agent-based systems. From the table, the competing agentic frameworks and prompting strategies include `ReAct`, `Reflexion`, `Self-Debugging`, `Self-Collaboration`, `MetaGPT`, `MapCoder`, `AgentCoder`, and `CodeCoR`(Brown et al., 2020; Huang et al., 2023b; Pan et al., 2025; Yao et al., 2023b; Shinn et al., 2023; Yao et al., 2023a; Hao et al., 2023; Zhang et al., 2023; Jiang et al., 2024; Chen et al., 2023; Dong et al., 2024; Wang et al., 2023; Li et al., 2025; Huang et al., 2023a; Islam et al., 2024).

**Evaluation Metrics** Following standard practice, we use the pass@k metric to evaluate code generation performance, where a solution is considered correct if it passes a set of unit tests. We primarily report pass@1 scores(Chen et al., 2021; Austin et al., 2021; Dong et al., 2025).

**Parameter Setting** For all experiments, we consistently applied an identical parameter set unless otherwise noted. The number of initial test cases was set to 15, and the number of distinct code snippets generated in each round was 20. We maintained a minimum pool of 20 test cases; if filtering processes reduced the number of test cases below this threshold (e.g., due to low scores), additional test cases were generated to meet this minimum. For iterative refinement, the number of codes chosen by acquisition function for further evaluation was set to 5.

### 3.2 EFFECTIVENESS IN AI FOR SCIENCE TASKS

We established our framework's general proficiency, which can be found in detail in the appendix, and can assert its up-to-SOTA level performance in general coding tasks. We can now investigate the framework's ability in scientific tasks by evaluating our framework's performance in the specialized and demanding domain of scientific code generation. We use two scientific code generation benchmarks to demonstrate its capabilities.

First, we assess our framework on the `SciCode` benchmark across a wide spectrum of base models, from the 1.7B parameter Qwen3 to powerful proprietary models like Claude-sonnet-4. The results, presented in Table 1, show that our framework provides a substantial and consistent performance uplift in all configurations. The gains are particularly striking for open-source models, with relative improvements of up to **87.1%** (for Qwen3-8b). Our framework enables smaller models to match the performance of significantly larger ones. For instance, in the 'Without Knowledge' case, Qwen3-14b with our framework achieves a 30.6 Resolve Rate on Subproblems, equaling the baseline of the Qwen3-235B-A22b-Instruct-2507, a model over **16 times** its size.

Second, to evaluate our framework on more complex, agentic workflows, we test it on the `ScienceAgentBench`, which involves more complex, multi-step scientific workflows, and we using GPT-4o as the base model. As shown in Table 2, our LCP framework achieves new state-of-the-art (SOTA) performance, particularly in the Valid Execution Rate (VER), where it scores **90.2%** (without knowledge) and **87.3%** (with knowledge), far surpassing all other methods. This exceptional execution success rate is critical for scientific applications, as it directly validates our framework's

core strength in producing robust, executable scientific code across diverse and complex application domains. This result, combined with leading scores in Success Rate (SR) and Code-Based Score (CBS), confirms our system's effectiveness in orchestrating the complex reasoning and execution steps essential for impactful AI4S applications.

Table 1: Model Performance Comparison on SciCode with various backbone models

| Model | Method | Without Knowledge | | With Knowledge | | Model | Method | Without Knowledge | | With Knowledge | |
|---|---|---|---|---|---|---|---|---|---|---|---|
| | | Sub (%) | Main (%) | Sub (%) | Main (%) | | | Sub (%) | Main (%) | Sub (%) | Main (%) |
| Qwen3-8b | Baseline | 13.2 | 0 | 19.8 | 1.5 | GPT-4o | Baseline | 24.1 | 1.5 | 33.7 | 7.7 |
| | Ours | 24.7(87.1%) | 4.6 | 27.4(38.4%) | 4.6 | | Ours | 37.2(54.3%) | 7.7 | 40.6(20.4%) | 10.8 |
| Qwen3-14b | Baseline | 17.7 | 1.5 | 25.0 | 6.2 | Deepseek-v3 | Baseline | 27.8 | 3.1 | 38.8 | 10.8 |
| | Ours | 30.6(72.9%) | 6.2 | 32.6(30.4%) | 6.2 | | Ours | 40.3(45.0%) | 10.8 | 42.4(9.28%) | 12.3 |
| Qwen3-32b | Baseline | 18.4 | 0 | 27.4 | 7.7 | Deepseek-R1 | Baseline | 29.6 | 4.6 | 37.8 | 10.8 |
| | Ours | 33.0(79.3%) | 6.2 | 36.1(31.8%) | 7.7 | | Ours | 41.0(38.5%) | 10.8 | 43.1(14.0%) | 13.8 |
| Qwen3-next-80b-a3b-instruct | Baseline | 21.5 | 3.1 | 32.6 | 12.3 | Claude-sonnet-4 | Baseline | 31.3 | 7.7 | 38.8 | 10.8 |
| | Ours | 37.5(74.4%) | 9.2 | 38.5(18.1%) | 10.8 | | Ours | 42.7(36.4%) | 13.8 | 43.8(12.9%) | 13.8 |
| Qwen3-235B-A22b-Instruct | Baseline | 30.6 | 4.6 | 37.2 | 10.8 | | | | | | |
| | Ours | 38.9(27.1%) | 9.2 | 41.0(10.2%) | 10.8 | | | | | | |

Table 2: Results on ScienceAgentBench using GPT-4o as base model with/without prior knowledge, compare with two different agent frameworks and baseline.

| Method | SR(w/o) | CBS(w/o) | VER(w/o) | SR(w/) | CBS(w/) | VER(w/) |
|---|---|---|---|---|---|---|
| Direct | 11.8 | 82.6 | 52.9 | 10.8 | 83.8 | 41.2 |
| OpenHands CodeAct | 19.6 | 83.1 | 78.4 | 27.5 | 86.3 | 73.5 |
| Self-Debug | 22.6 | 84.4 | 83.3 | 23.5 | 85.6 | 71.6 |
| LCP(Ours) | **26.5** | **85.1** | **90.2** | **27.5** | **86.4** | **87.3** |

## 3.3 ANALYSIS OF BAYESIAN RECURSIVE CO-UPDATING

In this section, we test our framework's core Bayesian iterative co-updating mechanism, validating that the Bayesian recursive co-updating strategy is effective at iteratively refining solutions. As illustrated across both general and scientific benchmarks in Figure 3, performance consistently and monotonically improves with an increasing number of iterations. On the general benchmarks (Figure 3a), the Pass@1 scores on `HumanEval` and `MBPP` show substantial gains in the first three iterations, with performance beginning to converge around the fourth or fifth iteration. This suggests an optimal balance between performance and computational cost. This same powerful trend is mirrored on the specialized `SciCode` benchmark (Figure 3b), where the performance score climbs steadily from a 27.1 to 37.2 after five iterations, demonstrating the broad applicability and success of our iterative refinement process.

We further analyze the components of this process by conducting an ablation study on the role of Adversarial Test Cases (ATC) within our LCP framework, as shown in Figure 3a. While the performance with and without ATC is comparable in the initial iterations, a clear divergence emerges from the third iteration onwards. The LCP framework augmented with ATC (dash-dot lines) consistently achieves higher Pass@1 accuracy across all metrics, underscoring the critical role of ATC. By dynamically challenging the generated code with difficult edge cases, the ATC mechanism compels the system to produce more robust and reliable solutions, validating it as a key driver of the performance gains observed in our co-updating loop.

## 3.4 ROBUSTNESS FOR NON-PROFESSIONAL USERS

Finally, we address robustness and accessibility to non-AI-professional science researcher by evaluating our framework's accessibility and effectiveness for users who may be domain experts but are not specialists in prompt engineering. To simulate this scenario, we compare the performance of both the baseline models and our framework under two conditions: one with a basic, un-optimized prompt ('Without Knowledge') and one with an expert-crafted prompt containing detailed domain knowledge ('With Knowledge'). The goal is to measure how sensitive each approach is to the quality of the initial prompt.

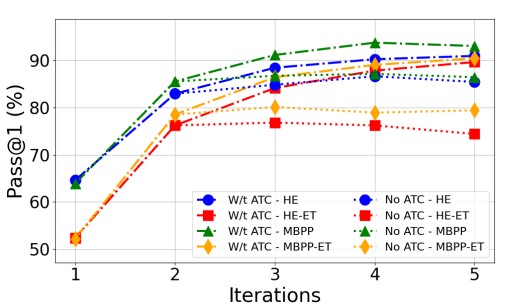

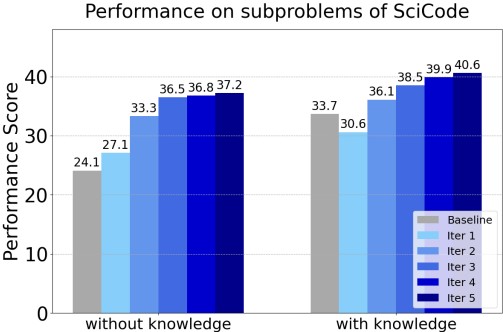

(a) Pass@1 of LCP with different iterations (GPT-3.5-turbo). Dash-dot and solid lines are LCP with and without ATC, respectively.

(b) Performance on SciCode Benchmark (GPT-4o) with different iteration numbers.

Figure 3: Illustration of LCP performance over: (a) different iteration number with and without ATC component on general code benchmark; (b) difficulty iteration number on the SciCode benchmark

The results, presented in Figure 4, clearly demonstrate our framework's superior robustness. The baseline models exhibit a large performance gap between the two conditions (represented by the shaded red area, 'Area (Baseline)'), indicating a strong dependency on expert prompting. In contrast, our framework significantly narrows this performance gap across the entire spectrum of models (represented by the much smaller shaded blue area, 'Area (Ours)'). This shows that our multi-agent system can internally elaborate on and refine basic instructions, compensating for the lack of initial detail. Most strikingly, a non-professional user with our framework ('Ours - Without Knowledge') consistently and substantially outperforms an expert user with the baseline model alone ('Baseline - With Knowledge').

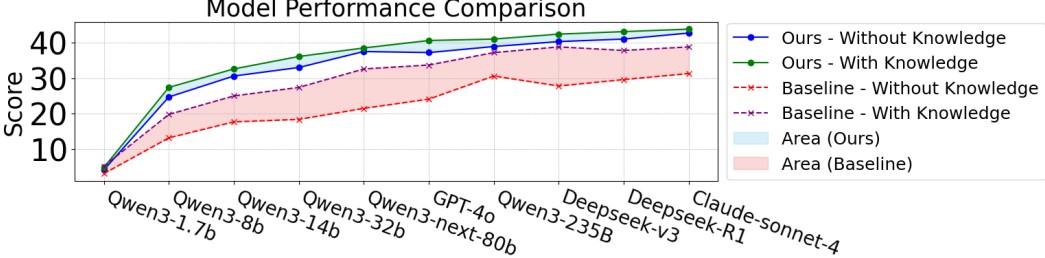

Figure 4: Model performance with basic vs. expert-crafted prompts. Our framework (blue/green lines) is significantly more robust to prompt quality than the baseline (red lines), showing a much smaller performance gap (shaded area) and achieving superior results even without expert knowledge.

## 4 CONCLUSIONS AND DISCUSSIONS

We propose a Bayesian adversarial multi-agent framework for AI-for-Science (AI4S) code generation that achieves state-of-the-art performance by iteratively refining prompt components through Bayesian updates. This approach mitigates cumulative error by treating tests and code with equivalent confidence, while an adversarial process guides a Task Manager (TM) agent to challenge a Solution Generator (SG) agent with progressively evolving tests. The framework's interactive planning scheme enables non-experts to translate vague prompts into validated workflows, effectively bridging the gap between AI-generated code and domain-specific needs. As demonstrated in Earth Science applications, our method helps democratize LLM tools for researchers without a technical background.

However, the framework has several limitations. First, its performance is dependent on the quality of the initial reference code, and it struggles to enforce implicit physical laws, which may require

future integration with symbolic verifiers. Second, the iterative adversarial-and-Bayesian refinement process consumes more tokens than one-shot/zero-shot prompting because it repeatedly generates and evaluates test cases, code candidates, and updated prompts. While this increased token budget can be a practical constraint for long pipelines, in reliability-critical scientific applications, we prioritize executable correctness and robustness over marginal token savings. Furthermore, evaluating the generated machine learning or deep learning models can be highly resource-intensive, and performance variability due to training dynamics and data poses an additional challenge to the update mechanism.

Future work will focus on extending the Bayesian updates to handle multi-modal inputs, such as equations and diagrams, and optimizing the iteration protocols for large-scale scientific simulations.

## ACKNOWLEDGMENTS

The computations in this research were performed using the CFFF platform of Fudan University.

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

## A  THE USE OF LARGE LANGUAGE MODELS (LLMS)

We confirm that LLMs were used for writing assistance and polishing of the manuscript, as well as editing Figure 2 based on our handmade version. They were not employed in the design of methods, implementation of experiments, or analysis of results.

## B  BAYESIAN OPTIMIZATION FOR CODE PERFORMANCE PREDICTION

For code embeddings, we use OpenAI's `text-embedding-3-large` model in the surrogate modeling pipeline. This embedding model is used to map generated code candidates into a semantic-structural vector space, after which the GP surrogate estimates candidate quality before expensive full execution.

In this embedding space, we can further obtain the pair-wise similarity between all the code embeddings. This similarity is computed using a squared exponential kernel:

$$k(\mathbf{x}_i, \mathbf{x}_j) = \exp\left(-\frac{d(\mathbf{x}_i, \mathbf{x}_j)^2}{2l^2}\right),$$

where $d(\mathbf{x}_i, \mathbf{x}_j)$ is the distance between the two code embeddings and $l$ is a length-scale parameter. These pairwise similarities $\{k(\mathbf{x}_i, \mathbf{x}_j)\}_{i,j}$ form the kernel matrix $\mathbf{K}$.

With the scores $S_2$ and the kernel matrix $\mathbf{K}$, we follow a standard Bayesian Optimization practice and fit a Gaussian Processes (GP) model. This model allows us to estimate the score of any new, *untested* code $\mathbf{x}_*$ without running the full evaluation. This also gives the 'likelihood' to guide the Bayesian update. For each new code, the trained GP provides a predictive distribution for its score, which is characterized by

- the mean function $\mu(\mathbf{x}_*)$ is the expected score of the new code conditioned on the tested codes:
$$\mu(\mathbf{x}_*) = \mathbf{k}_*^T(\mathbf{K} + \sigma_n^2\mathbf{I})^{-1}S_2.$$

- The variance function $\sigma^2(\mathbf{x}_*)$ represents our uncertainty about that predicted score:
$$\sigma^2(\mathbf{x}_*) = k(\mathbf{x}_*, \mathbf{x}_*) - \mathbf{k}_*^T(\mathbf{K} + \sigma_n^2\mathbf{I})^{-1}\mathbf{k}_*,$$

where $\mathbf{k}_*$ is the vector of kernel similarities between the new code $\mathbf{x}_*$ and all previously tested codes, and $\sigma_n^2$ is the noise term.

To decide which untested code to evaluate next, we employ a standard acquisition function that balances exploiting codes with high expected scores (exploitation) and exploring codes where the model is uncertain (exploration). We use the Upper Confidence Bound (UCB) acquisition function:

$$\text{UCB}(\mathbf{x}_*) = \mu(\mathbf{x}_*) + \kappa\sigma(\mathbf{x}_*).$$

The parameter $\kappa$ controls the trade-off between exploitation and exploration. The next code selected for full evaluation is the one that maximizes this UCB score:

$$\mathbf{x}_{\text{next}} = \arg\max_{\mathbf{x}_*} \text{UCB}(\mathbf{x}_*).$$

## C  PERFORMANCE ON GENERAL CODE GENERATION

To test our framework of general code generation, we establish our framework's proficiency on foundational code generation tasks. As detailed in Table 3, our framework(LCP) demonstrates a significant performance uplift across the `HumanEval`, `HumanEval-ET`, `MBPP`, and `MBPP-ET` benchmarks. When using GPT-3.5-Turbo as a backbone, LCP achieves pass@1 scores of **88.4%** on HumanEval and **91.1%** on MBPP, representing substantial relative improvements of **54.3%** and **74.5%** over the zero-shot baseline. This superior performance holds when using the more powerful

GPT-4 model, where LCP reaches **96.95%** on HumanEval, proving that our framework effectively enhances even the most capable foundation models. The consistent gains across all tests, especially the extended ('-ET') versions, validate the robustness and general applicability of our approach compared to other state-of-the-art agentic strategies.

Table 3: Pass@1 score comparison of various competing methods

| Models | HumanEval | HumanEval-ET | MBPP | MBPP-ET |
|---|---|---|---|---|
| Foundation Models (Zero-Shot) | | | | |
| Incoder (6.7B) | 15.2 | 11.6 | 17.6 | 14.3 |
| CodeLlama (34B) | 51.8 | - | 69.3 | - |
| GPT-3.5-turbo | 57.3 | 42.7 | 52.2 | 36.8 |
| Claude-instant-1 | 31.1 | 28.1 | 26.9 | 19.9 |
| GPT-4-turbo | 57.9 | 48.8 | 63.4 | 47.5 |
| GPT-4 | 67.6 | 50.6 | 68.3 | 52.2 |
| Agentic and Prompting Strategies (GPT-3.5-turbo) | | | | |
| Few-Shot | 67.7 (18.2%) | 54.9 (28.6%) | 65.8 (26.1%) | 48.3 (31.2%) |
| CoT | 44.6 (-22.2%) | 37.2 (-12.9%) | 46.1 (-11.7%) | 34.8 (-5.4%) |
| ReAct | 56.9 (-0.7%) | 49.4 (15.7%) | 67.0 (28.4%) | 45.9 (24.7%) |
| Reflexion | 68.1 (18.8%) | 50.6 (18.5%) | 70.0 (34.1%) | 47.5 (29.1%) |
| MapCoder | 80.5(40.5%) | 77.4(81.3%) | 78.9(51.1%) | 54.4(47.8%) |
| AgentCoder | 79.9(39.4%) | 77.4(81.3%) | 89.9(72.2%) | **89.1(142.1%)** |
| CodeCoR | 86.6(51.1%) | 80.5(88.5%) | 79.2(51.7%) | 65.2(77.2%) |
| **LCP (Ours)** | **88.4(54.3%)** | **84.1(97.0%)** | **91.1(74.5%)** | 86.4(134.8%) |
| Agentic and Prompting Strategies (GPT-4) | | | | |
| Reflexion | 91.0 (34.6%) | - | 77.1 (12.9%) | - |
| Self-Debugging | - | - | 80.6 (18.0%) | - |
| Self-Collaboration | 90.2 (33.4%) | 70.7 (39.7%) | 78.9 (15.5%) | 62.1 (19.0%) |
| MetaGPT | 85.9 (27.1%) | - | 87.7 (28.4%) | - |
| AgentCoder | 96.3(42.5%) | 86.0(70.0%) | 91.8(34.4%) | **91.8(75.9%)** |
| CodeCoR | 94.5(39.8%) | 83.5(65.0%) | - | - |
| **LCP (Ours)** | **96.95(43.4%)** | **88.41(74.7%)** | **92.51(35.4%)** | 89.70(71.8%) |

Furthermore, to test its capabilities on more complex problems, we evaluate our framework on the `APPS` benchmark against baseline and other two reference methods (LDB, LPW), categorized by task difficulty levels: Introductory, Interview, and Competition, using GPT-4o as the LLM backbone. Following existing literature, we switch our base model in the difficulty test for fair comparison. The results, illustrated in Figure 4, demonstrate the robust capabilities of our LCP framework. LCP consistently achieves the highest Pass@1 accuracy across all difficulty tiers, scoring 92.1% on Introductory tasks, 77.5% on Interview tasks, and a leading 38.0% on the challenging Competition tasks. This consistent superiority across varying complexities underscores the effectiveness of the LCP framework in generating correct solutions for a wide spectrum of programming challenges.

Table 4: Pass@1 accuracy of multi-agent framework (LPW) compared with baseline and LDB on APPS benchmark across different difficulty levels using GPT-4o as the LLM backbone.

| Difficulty Level | Baseline | LDB | LPW | LCP (Ours) |
|---|---|---|---|---|
| Introductory | 63.8 | 78.7 | 87.2 | 92.1 |
| Interview | 43.5 | 52.2 | 65.2 | 77.5 |
| Competition | 17.4 | 28.3 | 34.8 | 38.0 |

# D  ADDITIONAL CASE STUDY DETAILS

This section provides supplementary information for the case studies presented in the main paper, including detailed experimental setups, prompts, generated code, and further results. To guarantee a fair and interpretable comparison, we use GPT-4o as the unified base model across Cursor, Windsurf, and our framework during the case studies.

## D.1  CASE STUDY 1: BEACH PROFILE PREDICTION

### D.1.1  EXPERIMENTAL SETUP

The dataset utilized for this beach profile prediction study is organized into distinct training and testing sets, containing 536 and 242 rows respectively. Each data row represents a measurement point along

a beach profile, characterized by several key features: a numerical x coordinate denoting the distance from the profile's origin, serving as a primary input; a numerical y value representing the elevation at that distance, which is the target variable for prediction; and a categorical feature, "Dominant Wave Direction" (e.g., "ENE", "E"), necessitating encoding for model integration. Additional numerical columns are present, representing other relevant physical or environmental parameters that can be incorporated as supplementary features to enhance the predictive model's performance.

### D.1.2    USER'S PROMPT FOR BEACH PROFILE PREDICTION

Please refer to the Bruun model and the Dean model to build a mathematical model to discuss the change of sea level height with respect to the distance from the starting point, and build a deep learning model based on our data. The following is our data path and structure:

Data File Paths:

Training Data: " beach_profile_data / processed_data / beachdata_train . xlsx "
Test Data: " beach_profile_data / processed_data / beachdata_test . xlsx "
Data Structure Insights :
The training and testing datasets contain several columns. Key columns include:

x: A numerical column representing the distance from a profile 's origin; this is a primary input for predicting y.
y: A numerical column representing the elevation; this is the target variable for prediction .
Dominant Wave Direction: This is a categorical ( string ) type column (e.g., values like "ENE", "E", "NE"). This column will require appropriate encoding.
All other relevant columns you might select as features are expected to be in numerical ( integer or float ) format.

Create the code directly and The script should not run the main training or evaluation logic directly when the script file is executed. Instead, it should define all necessary functions with a main function main() that can be run WITHOUT ANY input. DO NOT USE tensorflow. Use provided data path in your code.

Listing 1: Prompt used for beach profile prediction.

### D.1.3    REFINED TASK DESCRIPTION BY LLM-TM AGENT

Please refer to the Bruun model and the Dean model to build a mathematical model to discuss the change of sea level height with respect to the distance from the starting point, and build a deep learning model based on our data. The following is our data path and structure:

Data File Paths:

Training Data: " beach_profile_data / processed_data / beachdata_train . xlsx "
Test Data: " beach_profile_data / processed_data / beachdata_test . xlsx "
Data Structure Insights :
The training and testing datasets contain several columns. Key columns include:

x: A numerical column representing the distance from a profile 's origin; this is a primary input for predicting y.
y: A numerical column representing the elevation; this is the target variable for prediction .
Dominant Wave Direction: This is a categorical ( string ) type column (e.g., values like "ENE", "E", "NE"). This column will require appropriate encoding.

All other relevant columns you might select as features are expected to be in numerical ( integer or float ) format.

create the code directly and The script should not run the main training or evaluation logic directly when the script file is executed. Instead, it should define all necessary functions.

=== Extra information ===

The Bruun model is a conceptual framework used in coastal engineering to predict shoreline retreat in response to sea-level rise. It assumes a balance of sediment in the cross-shore profile, resulting in both erosion and deposition occurring simultaneously as sea levels change. The formula often associated with the Bruun model states that the retreat distance (R) is proportional to the rise in sea level (S) divided by the height of the active profile (a):

$$R = \frac{S \times (d + B)}{a + B}$$

where:- $S$ is the sea-level rise.- $d$ is the depth of closure.- $B$ is the berm height.- $a$ is the active height.The Dean model focuses on sediment equilibrium profiles, often fitting the profile shape with the formula:

$$y = Ax^{2/3}$$

where:- $A$ is a parameter related to sediment characteristics.Both models serve to provide theoretical insights into the interaction of sea-level changes with coastal profiles.

=== Feature analysis ===

The dataset contains various features, including:
− **x**: Distance from a profile's origin, serving as a primary input.
− **y**: Elevation, the target variable for prediction.
− **Dominant Wave Direction**: A categorical feature indicating wave direction, requiring encoding.
− **Annual Mean Period, Annual Mean Spring Tidal Range, Annual Mean Tidal Range**: Numerical features capturing tidal metrics.
− **Breaker Wave Height Hb, Deep Water Wave Height Hd, Mean Wave Height**: Provide information about wave conditions.
− **Dimensionless Settling Velocity, High Tide Sediment Settling Velocity**: Indicative of sediment transport characteristics.
− **Latitude, Longitude**: Geographical coordinates potentially used for spatial analysis.
− **Mean Grain Size, Mean Grain Size (Mz)**: Describe sediment characteristics important for erosion/sediment transport modeling.
− **Skewness, Kurtosis, Sorting Coefficient**: Statistical measures about sediment distribution.

=== Extra Advice ===

To effectively use the provided features for the task, consider the following recommendations:

1. **Feature Engineering**:
   − Encode categorical variables such as "Dominant Wave Direction" using one−hot encoding or ordinal encoding.
   − Calculate additional derived features if necessary, such as interaction terms between wave heights and sediment properties.
   − Normalize or standardize numerical features to ensure they contribute equally to model optimization.

2. ∗∗ Incorporating Domain Knowledge∗∗:
   – Use the Bruun and Dean models to inform feature selection or crafting . Variables related to tide , wave height , and sediment characteristics are crucial .
   – Integrate expert knowledge about the relationships between features to inform model architecture in the neural network.

3. ∗∗Data Splitting and Validation ∗∗:
   – Ensure that the dataset is split appropriately into training and validation sets to evaluate model performance accurately .

4. ∗∗Model Architecture ∗∗:
   – Consider building a neural network that incorporates multiple hidden layers to capture complex interactions .
   – Use activation functions suitable for regression tasks , such as ReLU for hidden layers and linear activation for the output layer .

5. ∗∗Handling Geographical Variations ∗∗:
   – If spatial data relationships are significant , consider spatial analysis or geographically –weighted regression .

Listing 2: Refined task description by LLM-TM agent based on given prompt.(The highlight part is the additional prior knowledge generated by agent)

#### D.1.4 CODE GENERATED BY CURSOR (ONLY DATA PROCESS PART)

```python
def load_and_process_data(filepath, scaler=None, label_encoder=None,
    fit_scaler=False, fit_label_encoder=False):
    df = pd.read_excel(filepath)
    # Select features
    feature_cols = [col for col in df.columns if col not in ['y']]
    X = df[feature_cols].copy()
    y = df['y'].values.astype(np.float32)

    # Encode Dominant Wave Direction
    if 'Dominant Wave Direction' in X.columns:
        if label_encoder is None:
            label_encoder = LabelEncoder()
        if fit_label_encoder:
            X['Dominant Wave Direction'] = label_encoder.fit_transform(X[
    'Dominant Wave Direction'])
        else:
            X['Dominant Wave Direction'] = label_encoder.transform(X['
    Dominant Wave Direction'])
    else:
        label_encoder = None

    # Standardize numerical features
    num_cols = X.select_dtypes(include=[np.number]).columns.tolist()
    if scaler is None:
        scaler = StandardScaler()
    if fit_scaler:
        X[num_cols] = scaler.fit_transform(X[num_cols])
    else:
        X[num_cols] = scaler.transform(X[num_cols])

    return X.values.astype(np.float32), y, scaler, label_encoder, df['x'
    ].values.tolist(), df['y'].values.tolist()
```

Listing 3: Code for beach profile prediction generated by Cursor.

### D.1.5 CODE GENERATED BY WINDSURF (ONLY DATA PROCESS PART)

```python
1  # --- Data Loading and Preprocessing ---
2  def load_data(train_path, test_path):
3      train_df = pd.read_excel(train_path)
4      test_df = pd.read_excel(test_path)
5      # Select features
6      feature_cols = [col for col in train_df.columns if col not in ['y']]
7      target_col = 'y'
8      # Handle categorical encoding and scaling
9      scaler = StandardScaler()
10     label_encoders = {}
11     train_dataset = BeachProfileDataset(train_df, feature_cols,
       target_col, scaler=None, label_encoders=label_encoders, fit_scaler=
       True)
12     test_dataset = BeachProfileDataset(test_df, feature_cols, target_col,
        scaler=train_dataset.scaler, label_encoders=train_dataset.
       label_encoders, fit_scaler=False)
13     return train_dataset, test_dataset, feature_cols, label_encoders
```
Listing 4: Code for beach profile prediction generated by Windsurf.

### D.1.6 CODE GENERATED BY OUR FRAMEWORK (COMPLETE CODE)

```python
1  # Define the function to run the entire process
2  def run_beach_profile_model(train_data_path, test_data_path):
3      # Component 1: Data Loader
4      def data_loader(train_path, test_path):
5          # Load datasets
6          train_df = pd.read_excel(train_path)
7          test_df = pd.read_excel(test_path)
8
9          # One-hot encode categorical features
10         encoder = OneHotEncoder(sparse=False)
11         direction_encoded_train = encoder.fit_transform(train_df[['
       Dominant Wave Direction']])
12         direction_encoded_test = encoder.transform(test_df[['Dominant
       Wave Direction']])
13
14         # Drop the original categorical column and add the encoded
       columns
15         train_df = train_df.drop(columns=['Dominant Wave Direction'])
16         test_df = test_df.drop(columns=['Dominant Wave Direction'])
17         train_df = train_df.join(pd.DataFrame(direction_encoded_train,
       columns=encoder.get_feature_names_out(['Dominant Wave Direction'])))
18         test_df = test_df.join(pd.DataFrame(direction_encoded_test,
       columns=encoder.get_feature_names_out(['Dominant Wave Direction'])))
19
20         # Normalize numerical features
21         numerical_cols = train_df.select_dtypes(include=['int64', '
       float64']).columns
22         # Exclude variable 'y' and 'x'
23         numerical_cols = numerical_cols.drop(['y', 'x'])
24         scaler = StandardScaler()
25         train_df[numerical_cols] = scaler.fit_transform(train_df[
       numerical_cols])
26         test_df[numerical_cols] = scaler.transform(test_df[numerical_cols
       ])
27
28         return train_df, test_df
29
30     # Component 2: Theoretical Model Integration
31     def theoretical_model_integration(train_df, test_df):
32         # derived feature using Dean's model
33         train_df['Dean_Feature'] = train_df['x'] ** (2 / 3)
```

```python
34          test_df['Dean_Feature'] = test_df['x'] ** (2 / 3)

35
36          # Potential Bruun influence -> Using given example formula might
        require additional variables
37          train_df['Bruun_Feature'] = ((train_df['y'] * (train_df['Mean
        Wave Height'])) / (train_df['Mean Grain Size']))
38          test_df['Bruun_Feature'] = ((test_df['y'] * (test_df['Mean Wave
        Height'])) / (test_df['Mean Grain Size']))

39
40          return train_df, test_df

41
42      # Component 3: Deep Learning Model
43      def deep_learning_model(X_train, y_train):
44          # Define the model architecture
45          class SimpleNN(nn.Module):
46              def __init__(self, input_size):
47                  super(SimpleNN, self).__init__()
48                  self.layer1 = nn.Linear(input_size, 128)
49                  self.layer2 = nn.Linear(128, 64)
50                  self.layer3 = nn.Linear(64, 32)
51                  self.output = nn.Linear(32, 1)
52                  self.relu = nn.ReLU()

53
54              def forward(self, x):
55                  x = self.relu(self.layer1(x))
56                  x = self.relu(self.layer2(x))
57                  x = self.relu(self.layer3(x))
58                  x = self.output(x)
59                  return x

60
61          # Initialize and compile the model
62          model = SimpleNN(X_train.shape[1])
63          return model

64
65      # Component 4: Model Training and Evaluation
66      def model_training_evaluation(model, train_df, test_df):
67          # Split the data into features and target
68          X_train = train_df.drop(columns=['y']).to_numpy()
69          y_train = train_df['y'].to_numpy()

70
71          X_test = test_df.drop(columns=['y']).to_numpy()
72          y_test = test_df['y'].to_numpy()

73
74          # Convert arrays to PyTorch tensors
75          X_train_tensor = torch.tensor(X_train, dtype=torch.float32)
76          y_train_tensor = torch.tensor(y_train, dtype=torch.float32).view
        (-1, 1)

77
78          X_test_tensor = torch.tensor(X_test, dtype=torch.float32)
79          y_test_tensor = torch.tensor(y_test, dtype=torch.float32).view
        (-1, 1)

80
81          # Create dataloaders
82          train_dataset = TensorDataset(X_train_tensor, y_train_tensor)
83          train_loader = DataLoader(train_dataset, batch_size=32, shuffle=
        True)

84
85          # Define loss function and optimizer
86          criterion = nn.MSELoss()
87          optimizer = optim.Adam(model.parameters(), lr=0.001)

88
89          # Training loop
90          model.train()
91          for epoch in range(100):
92              for batch_X, batch_y in train_loader:
```

```
93                  optimizer.zero_grad()
94                  outputs = model(batch_X)
95                  loss = criterion(outputs, batch_y)
96                  loss.backward()
97                  optimizer.step()
98
99          # Evaluation metrics
100         model.eval()
101         with torch.no_grad():
102             predictions = model(X_test_tensor)
103             mse = criterion(predictions, y_test_tensor).item()
104             rmse = np.sqrt(mse)
105             mae = np.mean(np.abs(predictions.numpy() - y_test_tensor.
    numpy()))
106             r_squared = 1 - (np.sum(np.square(predictions.numpy() -
    y_test_tensor.numpy())) / np.sum(np.square(y_test_tensor.numpy() - np
    .mean(y_test_tensor.numpy()))))
107
108         return {'RMSE': rmse, 'MAE': mae, 'R2': r_squared}
109
110     # Execute the process
111     train_df, test_df = data_loader(train_data_path, test_data_path)
112     train_df, test_df = theoretical_model_integration(train_df, test_df)
113     train_df = train_df.dropna()
114     test_df = test_df.dropna()
115     model = deep_learning_model(train_df.drop(columns=['y']).to_numpy(),
    train_df['y'].to_numpy())
116     evaluation_results = model_training_evaluation(model, train_df,
    test_df)
117
118     return evaluation_results
```

Listing 5: Code for beach profile prediction generated by Our Framework.

### D.1.7 RESULTS AND DISCUSSION

In this case study focused on beach profile prediction, the primary objective was to integrate established theoretical models—specifically the Bruun and Dean models—with a deep learning methodology, as per the user's explicit requirement. An examination of the approaches reveals that while both the Cursor and Windsurf frameworks implemented standard data preprocessing techniques such as numerical feature standardization and one-hot encoding for categorical data, they did not incorporate the specified theoretical models. In contrast, our LCP framework successfully addressed the user's need by calculating additional derived features based on the Bruun and Dean models. This direct integration of domain-specific theoretical knowledge into the feature set represents a key differentiator in our approach.

The impact of this tailored feature engineering is reflected in the prediction performance, as illustrated in 5. The results indicate that the LCP framework yielded predictions superior to those generated by Cursor. Furthermore, LCP's predictive accuracy was observed to be closely comparable to the results from Windsurf. This suggests that the inclusion of theoretically-derived features not only fulfilled a critical user requirement but also contributed positively to the model's ability to accurately predict beach profile changes, positioning LCP as a more comprehensive solution for this specific task.

### D.2 CASE STUDY 2: BRAIN MRI SEGMENTATION

### D.2.1 EXPERIMENTAL SETUP (BRAIN MRI)

This study utilizes a subset of the "LGG MRI Segmentation" dataset, which contains brain Magnetic Resonance Images (MRI) and corresponding manual FLAIR abnormality segmentation masks for patients with Lower Grade Glioma (LGG). The original dataset, sourced from The Cancer Imaging Archive (TCIA), includes data from 110 patients. For this experiment, data from 30% of these patients was selected. This selected patient data, comprising MRI slices (typically 256x256 pixels)

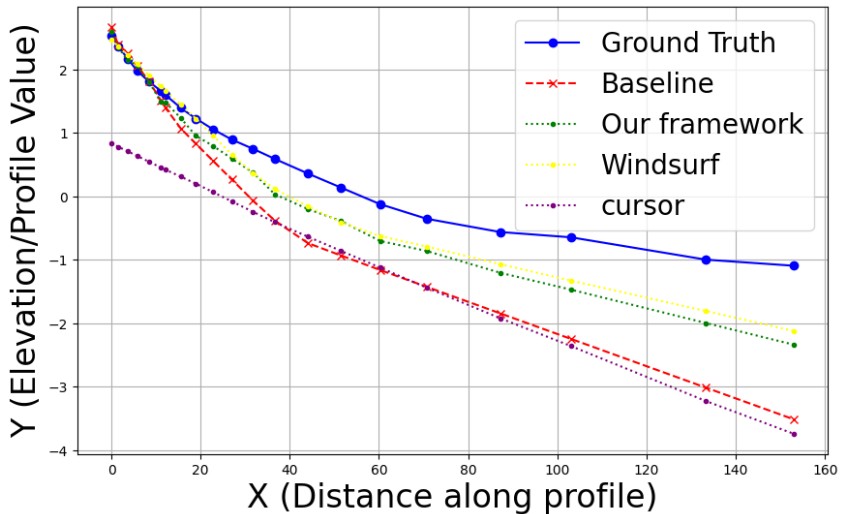

Figure 5: Beach profile prediction results comparison.

and their associated segmentation masks, was then partitioned into training, validation, and testing sets with a 7:1.5:1.5 ratio, respectively, to develop and evaluate a deep learning model for brain tumor segmentation.

### D.2.2 Prompt for Brain MRI Segmentation

```
Develop an image segmentation model to accurately and efficiently  identify and segment
FLAIR abnormalities in  brain  MR images from the LGG Segmentation Dataset.

Dataset  Format & Structure :

Base Path :  r"E:\python_project_new\brain_case_study \lgg−mri−segmentation\case_study"

 Organization :
The base  path  contains  train /,  val /,  and test /  subfolders .
Each of  these  subfolders  contains  two further  subfolders :  images/ and masks/.

Image Files  (images/):
Format: 3−channel . tif   files .
Naming: TCGA_<institution−code>_<patient−id>_<slice−number>.tif.

Mask Files  (masks/):
Format: Binary,  1−channel . tif   files .
Naming: TCGA_<institution−code>_<patient−id>_<slice−number>_mask.tif.
```

Listing 6: User's Prompt used for Brain MRI segmentation.

### D.2.3 Refined Prompt by LLM-TM agent

```
=== Role ===
 ...
=== Extra  Advice ===

− ∗∗Data Preprocessing :∗∗ Normalize image  intensity   values  (min−max or z−score
 normalization ). Ensure all  images are  resized /cropped to  the  same shape (commonly
```

256x256 or as appropriate for U−Net input). If channel information is redundant, you could reduce it to one channel for efficiency.
− ∗∗Label Preparation:∗∗ Ensure mask values are binary (0 and 1). Double−check that mask and image align correctly for every slice.
− ∗∗Data Augmentation:∗∗ To combat overfitting and improve generalization, apply transformations such as flipping, rotation, scaling, and intensity shifts, ensuring they are applied equally to images and masks.
− ∗∗Model Choice:∗∗ Use U−Net (or a lightweight modification thereof) due to its strong performance in medical image segmentation with relatively small datasets.
− ∗∗Loss Function:∗∗ Use a combination of Binary Cross Entropy and Dice Loss to penalize both pixel−wise errors and poor overlap.
− ∗∗Batching:∗∗ Due to likely high resolution and small dataset, use small batch sizes and possibly gradient accumulation.
− ∗∗Evaluation:∗∗ Report Dice on the test set (already specified). Consider visualizing a few results for qualitative assessment, e.g., overlay masks or compare ground truth to prediction.
− ∗∗ Visualization :∗∗ Save (or plot) a grid of test images, ground truth, and predictions for several slices to inspect the model's performance.

By thoroughly preparing the dataset, using appropriate augmentations, and selecting a suitable architecture and losses, you can build a robust FLAIR abnormality segmentation model for the LGG dataset. Ensure reproducibility by setting random seeds and keeping all data splits strictly separate during model development and evaluation.
...

Listing 7: Partial Refined Prompt for Brain MRI segmentation, generated by LLM-TM agent

### D.2.4 CODE GENERATED BY CURSOR (ONLY MAIN TRAINING LOOP)

```
...
# --- Training Loop ---
def train_one_epoch(model, loader, optimizer, criterion, device):
    model.train()
    epoch_loss = 0
    for imgs, masks in tqdm(loader, desc='Train', leave=False):
        imgs, masks = imgs.to(device), masks.to(device)
        optimizer.zero_grad()
        outputs = model(imgs)
        loss = criterion(outputs, masks)
        loss.backward()
        optimizer.step()
        epoch_loss += loss.item() * imgs.size(0)
    return epoch_loss / len(loader.dataset)

# --- Main Function ---
def main():
    device = torch.device('cuda' if torch.cuda.is_available() else 'cpu')
    # Datasets
    ...
    # Model
    model = UNet().to(device)
    optimizer = optim.Adam(model.parameters(), lr=1e-3)
    criterion = nn.BCELoss()
    # Training
    best_val = 0
    for epoch in range(1, 11):  # 10 epochs
        train_loss = train_one_epoch(model, train_loader, optimizer,
    criterion, device)
        val_dice = evaluate(model, val_loader, device)
        if val_dice > best_val:
```

```
31              best_val = val_dice
32              best_model = model.state_dict()
33      # Test
34      model.load_state_dict(best_model)
35      test_dice = evaluate(model, test_loader, device)
36      return model, test_loader, device, test_dice
37      ...
```

Listing 8: Code for Brain MRI segmentation generated by Cursor.

### D.2.5 CODE GENERATED BY WINDSURF (ONLY MAIN TRAINING LOOP)

```
1  ...
2  def train_one_epoch(model, loader, optimizer, criterion):
3      model.train()
4      epoch_loss = 0
5      for imgs, masks in loader:
6          imgs, masks = imgs.to(DEVICE), masks.to(DEVICE)
7          optimizer.zero_grad()
8          outputs = model(imgs)
9          loss = criterion(outputs, masks)
10         loss.backward()
11         optimizer.step()
12         epoch_loss += loss.item() * imgs.size(0)
13     return epoch_loss / len(loader.dataset)
14
15 def main():
16     base_path = ...
17     ...
18     model = UNet().to(DEVICE)
19     optimizer = optim.Adam(model.parameters(), lr=1e-3)
20     criterion = nn.BCELoss()
21     best_val_dice = 0
22     for epoch in range(1, 11):  # 10 epochs
23         train_loss = train_one_epoch(model, train_loader, optimizer,
       criterion)
24         val_dice = evaluate(model, val_loader)
25         if val_dice > best_val_dice:
26             best_val_dice = val_dice
27             best_model_state = model.state_dict()
28     model.load_state_dict(best_model_state)
29     test_dice = evaluate(model, test_loader)
30     return model, test_dice
31 ...
```

Listing 9: Code for Brain MRI segmentation generated by Windsurf.

### D.2.6 CODE GENERATED BY OUR FRAMEWORK (ONLY MAIN TRAINING LOOP)

```
1  ...
2  class DiceLoss(nn.Module):
3      def __init__(self, eps=1e-7):
4          super().__init__()
5          self.eps = eps
6      def forward(self, logits, targets):
7          inputs = torch.sigmoid(logits)
8          targets = (targets > 0.5).float()
9          intersection = (inputs * targets).sum(dim=(2,3))
10         unionset = inputs.sum(dim=(2,3)) + targets.sum(dim=(2,3))
11         dice = (2 * intersection + self.eps) / (unionset + self.eps)
12         loss = 1 - dice
13         return loss.mean()
14
15 # ------- Train and Evaluate --------
```

```python
16  def train_model(
17      model,
18      dataloader_dict,
19      device,
20      n_epochs=10,
21      lr=1e-3,
22      patience=2,
23  ):
24      optimizer = torch.optim.Adam(model.parameters(), lr=lr)
25      bce_loss = nn.BCEWithLogitsLoss()
26      dice_loss = DiceLoss()
27      best_state = copy.deepcopy(model.state_dict())
28      best_val_dice = -np.inf
29      epochs_no_improve = 0
30      for ep in range(n_epochs):
31          model.train()
32          train_loss = []
33          train_dice = []
34          for x, y in dataloader_dict['train']:
35              x = x.to(device)
36              y = y.to(device)
37              optimizer.zero_grad()
38              pred = model(x)
39              loss = 0.5*bce_loss(pred, y) + 0.5*dice_loss(pred, y)
40              loss.backward()
41              optimizer.step()
42              train_loss.append(loss.item())
43              train_dice.append(dice_coefficient(pred, y))
44          # Validation
45          ...
46          # Early stopping
47          ...
48      model.load_state_dict(best_state)
49      return model
50  ...
51  # ------------- MAIN PIPELINE ---------------
52  def main():
53      """
54      Loads data, trains segmentation model, returns model and average Dice
       on test set.
55      """
56      # ---- Settings (tuned for quick/robust test) ----
57      DATA_PATH = ...
58      ...
59      # ---- Model ----
60      model = UNetLite(in_ch=3, out_ch=1).to(device)
61      # ---- Train ----
62      model = train_model(
63          model,
64          dataloader_dict,
65          device,
66          n_epochs=N_EPOCHS,
67          lr=LR,
68          patience=PATIENCE
69      )
70      # ---- Evaluate test set ----
71      test_dice = evaluate_model(model, dataloader_dict['test'], device)
72      return model, test_dice
```

Listing 10: Code for Brain MRI segmentation generated by Our Framework.

### D.2.7   RESULTS AND DISCUSSION (BRAIN MRI)

In this case study, the objective was to develop a model for accurate and efficient identification and segmentation of abnormalities in brain MR images, based on a relatively open-ended user description that primarily specified the data format structure. While both the Cursor and Windsurf frameworks opted for a traditional U-Net architecture and utilized BCELoss for model training, our LCP framework adopted a different strategy. LCP generated code implementing a "Lite U-Net," a more streamlined architecture designed for faster computation. Furthermore, for the training process, LCP combined nn.BCEWithLogitsLoss() with a custom DiceLoss().

The practical outcomes of these differing approaches are evident in the performance metrics presented in 5. Most notably, the LCP framework demonstrated a significant advantage in computational efficiency, with its generated code requiring only approximately one-quarter of the training time compared to the solutions from Cursor and Windsurf. In terms of segmentation accuracy, the LCP framework achieved a Dice score on the test dataset that surpassed Cursor's results and was only marginally lower than that of Windsurf. This indicates that LCP's choice of a lighter model and a compound loss function provided a highly efficient solution that maintained a competitive level of accuracy, effectively addressing the user's call for both efficiency and accuracy in a complex image segmentation task.

| Framework | Cursor | Windsurf | LCP (ours) |
|---|---|---|---|
| Dice Score | 0.6627 | **0.7232** | 0.7185 |
| Training time | 127.9s | 131.6s | **36.2s** |

Table 5: Comparison of Dice scores on test data with code generated by different frameworks.

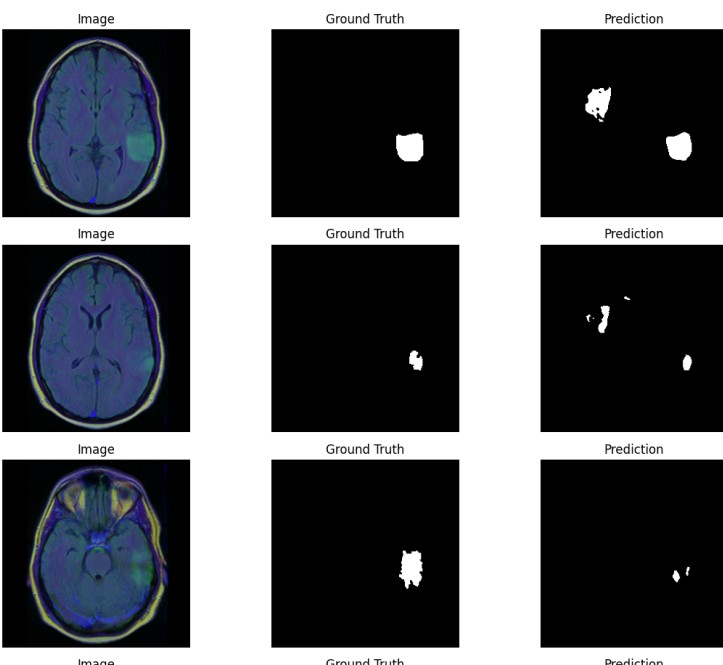

Figure 6: Brain MRI Segmentation by Cursor.

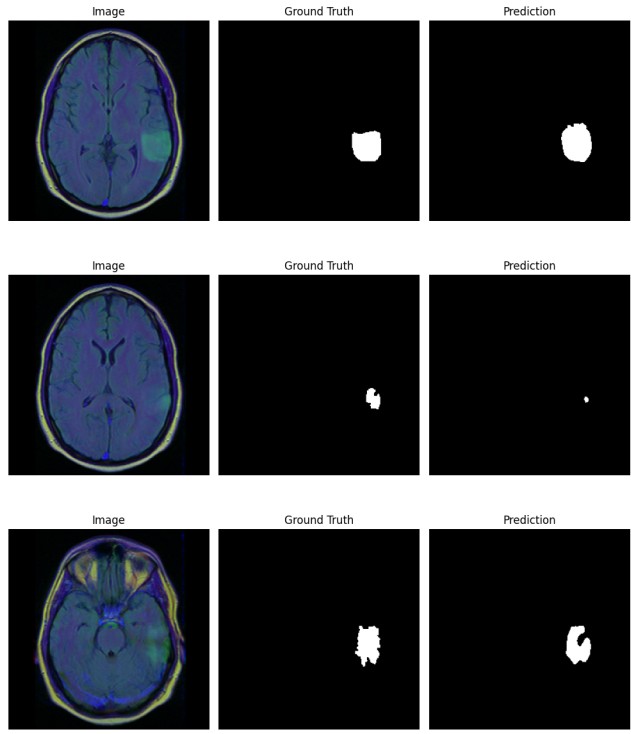

Figure 7: Brain MRI Segmentation by Windsurf.

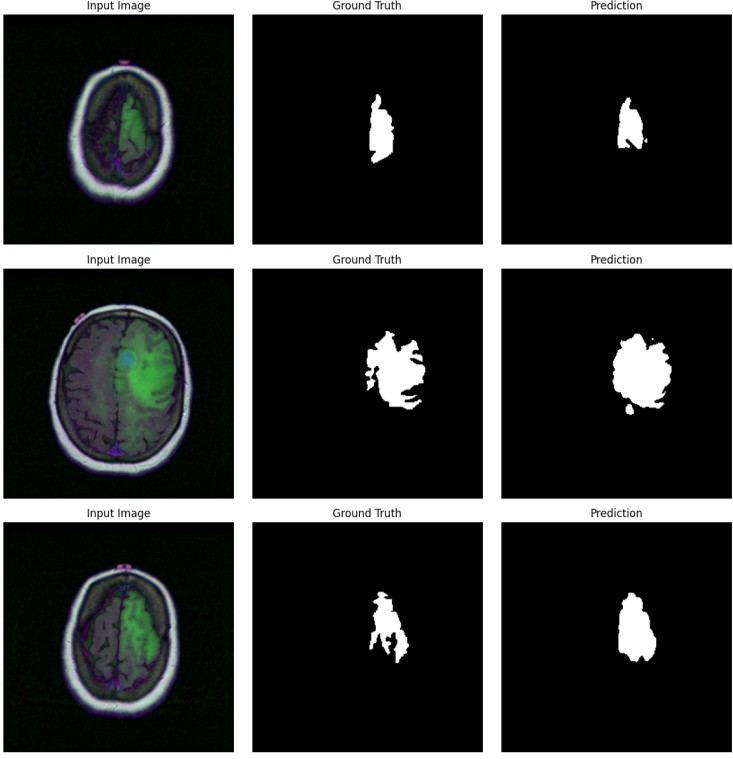

Figure 8: Brain MRI Segmentation by LCP.

### D.3 EXPERT ADVICE FILTERING AND LEAKAGE PREVENTION

In our framework, expert advice is restricted to domain knowledge such as physical formulas, data constraints, and high-level scientific logic (e.g., guidance like "use the Bruun model equation"), rather than implementation-level solution code. In our experiments, this advice was strictly filtered to avoid ground-truth code leakage. In addition, except SciCode (With Knowledge) and the targeted case-study settings, we did not inject expert knowledge, reducing leakage risk and preserving fair comparisons.

### D.4 TRAINING PERFORMANCE

To train our model, we collected 120 unique answers for a specific LeetCode problem. This dataset was then divided, with 60 answers designated for training. The remaining 60 answers from the target problem were combined with 60 code samples from different LeetCode problems to form our test dataset.

The training process demonstrated efficient learning, as illustrated in 9, which shows the training loss plotted against epochs. The loss converged rapidly, showing a significant decrease from the start to the 10th epoch.

Upon evaluating the trained model on the test dataset, using a classification threshold of 0, we achieved promising results. The model demonstrated an Accuracy of 0.8487, a ROC AUC score of 0.9322, and Precision, Recall, and F1-score all at 0.8475.

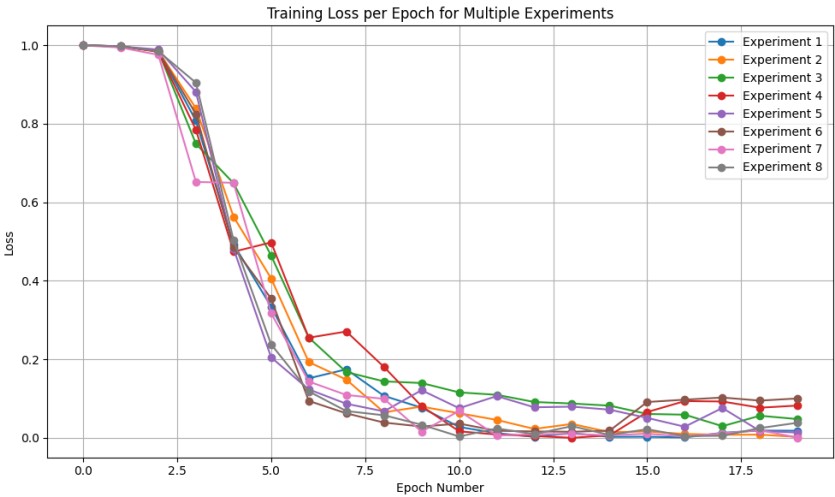

Figure 9: Training loss against epoch for the IRL experiment.

### D.5 STATE AND ACTION SPACE DEFINITION EXAMPLE

```
Vocabulary:
< state ><UNK>: 0
Name_U1: 1
 ...
Name_<UNKNOWN>: 11
arg_U1: 12
 ...
arg_<UNKNOWN>: 22
Assign_U1: 23
 ...
Assign_<UNKNOWN>: 33
< state >Assign(Assign_U1): 34
< state > Attribute ( Attribute ): 35
```

```
<state>ClassDef(ClassDef): 36
<state>FunctionDef(FunctionDef): 37
<state>Import(math): 38
<state>ImportFrom(os::path): 39
<state>arg(arg_U1): 40
 ...
Add: 43
BinOp: 44
Call(sqrt): 45
Constant: 46
Load: 47
Module: 48
Mult: 49
 ...
```

Listing 11: State and Action Space Definition.

Based on the State and Action Space defined above, we will transfer code into a state-action trajectory, for example:

```
Code:
import math
from os import path
x = math.sqrt(16)

It would be converted into:
State:
[[38], [38, 39], [34, 35, 38, 39]]
Action:
[[38], [39], [34, 35, 45, 46, 50, 51]]
```

Listing 12: Single example about code processing

## E  PROMPTS USED FOR LARGE LANGUAGE MODELS (LLMS)

This section details the specific prompts provided to the Large Language Models (LLMs) for various sub-tasks within our framework. These prompts are displayed to mimic structured textual input.

### E.1  LLM-TM (TASK MANAGER) PROMPTS

#### E.1.1  PROMPT FOR DATA ANALYSIS AND PRIOR KNOWLEDGE REFINEMENT

```
I am working on a code generation task:
<task_description_start>
{task_description}
<task_description_end>

I need you to analyze the data I will be working with and add extra information about
the task escription. Please structure your response as follows:

<extra_information>
Based on the task description, provide any additional information or context that might
be relevant to the task. For example, adding mathematical formulas, domain−specific
knowledge, or any other relevant information that can aid in the code generation
process.
```

```
</ extra_information >

< feature_analysis >
Provide a brief introduction to the features present in the data.
If specific feature information is not available, please describe the overall data
 structure you would expect or require for this task.
</ feature_analysis >

<advice>
Based on the task description, extra knowledge and the data features (or expected data
 structure),
provide advice on how these features can be effectively used to accomplish the project,
and how to incorporate the additional information into the project.
Suggest potential data transformations, feature engineering steps, or specific ways to
leverage features and extra domain-specific knowledge in the code generation process.
</advice>
```

Listing 13: LLM-TM: Prompt for data analysis.

```
I have an initial task description and some analysis regarding the data and approach.
I need you to synthesize all this information into a single, clear, and more detailed
 refined task description.

The refined task description should incorporate insights from the feature analysis and
the advice provided, making it more actionable and comprehensive for a code generation
system. It should clearly state the goal, the expected inputs (data/ features), and any
key steps or considerations mentioned in the advice.

Here is the information:

< original_task_description_start >
{ original_task_description }
< original_task_description_end >

< extra_information_start >
{ extra_information }
<extra_information_end>

< feature_analysis_start >
{ feature_analysis }
< feature_analysis_end >

< advice_start >
{advice}
<advice_end>

Now, please provide the refined task description. Output only the refined task
 description itself, without any extra conversational text or tags.
Refined Task Description:
```

Listing 14: LLM-TM: Prompt for prior knowledge refinement.

### E.1.2 PROMPT FOR TASK DECOMPOSITION

You are an expert agent specialized in decomposing code generation tasks into structured, detailed, and clear subtasks and then give a detailed overall plan based on your defined subtasks. Given a simple high−level task description, your job is to break it down into logical subtasks that clearly illustrate the workflow and ensure easy understanding and execution.

Each decomposed subtask should aim to create a function or class as a reusable component contributing to the overall task. If the provided task is too simple or atomic to require multiple components, your decomposition should only contain a single component.

Your output must strictly follow the format below:

```
<components>
{
  "component_1": {
    " step_task_description ": str ,
    "input_format": [[ type, shape or null ]],
    "output_format": [[ type, shape or null ]],
    "work_flow": [ str ],
    " test_case_generation_advise ": [ str ]
  },
  "component_2": {
    " step_task_description ": str ,
    "input_format": [["type", shape or null ]],
    "output_format": [["type", shape or null ]],
    "work_flow": [ str ],
    " test_case_generation_advise ": [ str ]
  },
   ...
}
</components>

< overall_plan >
{
  "input_format": [["type", shape or null ]],
  "output_format": [["type", shape or null ]],
  "components": [ str ],
  "plan": [ str ],
  " test_case_generation_advise ": [ str ]
}
</ overall_plan >
```

Here are additional detailed explanations of each field :

For <components>:
− ∗∗component_X∗∗: The key represents the subtask name, it should be replaced by the actual class / function name of the component (e.g., "merge_arrays", "calculate_median ") .
− ∗∗ step_task_description ∗∗: Provide a clear and concise description of exactly what this subtask aims to achieve, specifically mentioning the intended functionality or role of the created component (function / class ).
− ∗∗input_format∗∗: Describe the format of each input argument required for this subtask. It is a list of lists , where each inner list has two elements:
  − The first element indicates the data type (e.g., " list ", " dict ", NumPy array, torch . Tensor). DO make sure the data type is a string .
  − The second element indicates the fixed shape if applicable ; otherwise, it is null .
− ∗∗output_format∗∗: Describe the format of each output argument generated by this subtask. It follows the same list structure as ' input_format ', note that it has to be a list of lists .

− **work_flow**: Provide a detailed step−by−step plan that outlines the workflow of how the component functions to achieve the subtask.
− ** test_case_generation_advise **: Provide a list of detailed guidelines or suggestions aimed at generating diverse and comprehensive test cases, explicitly mentioning potential edge cases and critical scenarios that need coverage.

For <overall_plan>:
− **input_format**: Describe the format of the input arguments required for the overall task. It follows the same structure as 'input_format' in the component section.
− **output_format**: Describe the format of the output arguments generated by the overall task. It follows the same structure as 'output_format' in the component section.
− **components**: List the components in the order.
− **plan**: Provide a detailed step−by−step plan that outlines the workflow of how the components interact with each other to achieve the overall task. This should be a high−level description of the process.
− ** test_case_generation_advise **: Provide a list of detailed guidelines or suggestions aimed at generating diverse and comprehensive test cases for the overall task, explicitly mentioning potential edge cases and critical scenarios that need coverage.

Your decomposition should strive for clarity, correctness, modularity, and ensure each step can be tested independently. Now, given the following simple task description:

"{{TASK_DESCRIPTION}}"

Use <> to indicate both start and end of the component part and the overall plan. Ensure that the components and the overall plan are clearly separated.

Please provide your structured decomposition according to the instructions above.

Listing 15: LLM-TM: Prompt for task decomposition.

You are an expert agent specialized in refining and improving code generation plans through iterative feedback. Given a task description, previous decomposition output, and user feedback, your job is to critically analyze the existing plan and modify it accordingly while maintaining the required output format.

Carefully review the previous components and overall plan, then:
1. Preserve correct / valid elements that don't conflict with the feedback
2. Make targeted modifications based on the user's specific advice
3. Ensure consistency between components and overall plan
4. Verify input / output formats and workflow logic
5. Check for any introduced errors during modification

The input consists of three elements:
− Original Task Description: "{{TASK_DESCRIPTION}}"
− Previous Decomposition Output:
{{PREVIOUS_OUTPUT}}
− User Feedback: "{{USER_ADVICE}}"

Your output must STRICTLY follow the original format with these sections:
<components>...</components>
<overall_plan>...</overall_plan>

Follow these guidelines:
− Explicitly address all points in the user feedback
− Clearly document any changes made from previous version
− Preserve JSON structure and formatting requirements
− If feedback contradicts original requirements, prioritize feedback

Again, user feedback is: "{{USER_ADVICE}}"

Provide your refined decomposition with clear explanations of changes in the component descriptions and overall plan. Ensure modularity, testability, and coverage of edge cases mentioned in feedback.

Listing 16: LLM-TM: Prompt for plan refinement based on user feedback.

### E.1.3 PROMPT FOR TEST CASES GENERATION

You are a test case generation agent. Your task is to create Python test functions to validate a code generation task based on the provided specifications. Follow these instructions carefully:

### Input Specifications:
- **Task Description**:
{ task_descr_str }
- **Input Format**:
{ input_descr_str }
- **Output Format**:
{ output_descr_str }
- **Components Used**: {components_str}
- **Plan**:
{ plan_str }
- **Test Case Advise**:
{ advisory_list }

### Requirements:
1. **Test Function Structure**:
    - Each test function must accept **only the function under test** as its parameter (e.g., 'def test_case(func):...') .
    - Return 'True' if the test passes, 'False' otherwise. Do not use assertions, please return a boolean value.
    - Include input generation, runtime checks, code inspection, or result validation within the function.

2. **Test Types** (use one of these for indicating the test_type):
    - 'correctness': Validate output against expected results for specific inputs.
    - 'edge_case': Test inputs like empty lists, extreme values, or invalid data.
    - 'runtime': Measure execution time (e.g., ensure it's below a threshold).
    - 'component_check': Verify the function's code uses specified components (e.g., via string inspection).
    - 'error_handling': Check if errors are raised for invalid inputs.

3. **Test Case Diversity**:
    - Cover all provided advisories.
    - Include at least one test per advisory and one for each test type where applicable.

### Output Format:
For each test case, you need to firstly define the Test Types to indicate what type of test case you are going to create and then give the reasoning and explanation of the test case. After that, generate the test function based on the your reasoning.

For each test function, return with following structure:

<Type>

```
Pick one of  correctness |edge_case|runtime|component_check|error_handling
</Type>
<Planning>
Introduce how would you design the  test  function . Specify the purpose of the  test
function  and the  reasoning behind it . Explain step by step why your test  case is
correct  and what is  the  expected  output .
</Planning>

def  test_case (func):
    # Your test  function  code here
     return  True  or False  as  test  result , and a message


If you are going to create  multiple  test  cases , please  separate  them with <separator>
tag .

{example_text}
Generate  test  cases  that  rigorously  validate  the  function's  behavior , code  structure ,
and performance.
You MUST strictly follow  the  output  format and  structure . The generated  test  functions
MUST be runnable function  that  use  another  python  function  as  its  parameter  and it
should  output  both  the  Test  result (True  or False ) and a message  to  give  extra
information  about  the  test  result .( For example, f"Test  failed : expected X but got  Y" or
"Test  failed : output  with  shape  [x1, y1]  but  got  [x2, y2]", where the  X, Y and shapes
need  to  be  replaced  by  the  actual  output  and expected  output  in  test  function ).
```

Listing 17: LLM-TM: Prompt for test case generation.

### E.2    LLM-CG (CODE GENERATION) PROMPTS

#### E.2.1    PROMPT FOR CODE GENERATION

```
=== Role ===
You are a highly  skilled  coding  assistant  designed  to generate  clear ,  efficient , and
 correct  code based on  structured  task  descriptions  and detailed  plans  provided  by the
user . Your responses  must precisely  follow  the  instructions , formats , and constraints
given by the  user , and you must  strictly  adhere  to input−output  formats , workflows, and
 specific  guidelines  outlined .

=== Task Description  ===
{ task_description }

=== Components ===
{components_description}

=== Overall  Plan  ===
{ plan_text }

=== Test  Cases  ===
{ sampled_test_cases }

=== Instructions  ===
Generate  the  COMPLETE code based on the components and plan above.
DO MAKE SURE the complete code is a runnable function,  all  components are  correctly
 integrated  with in  this  function .
The complete  function  should  take  the  input  arguments  as  specified  in  the  overall  plan
and return  the  output  as  specified .
```

Please add as much comments as possible to your code to explain the logic and any
 critical steps .
 Structure your response as follows :

Your code here . DO make sure the output is a single function that integrates all
components.

<Planning>
A detailed step−by−step explanation of the code's workflow.
</Planning>
<Main Function Name>
The name of the main function that integrates all components.
</Main Function Name>
Provide the code with the same indicator and structure as shown in Instructions . DO
NOT return any test cases or example usages in your code!

Listing 18: LLM-CG: Prompt for code generation.

### E.2.2 PROMPT FOR CODE REFINEMENT

=== Role ===
You are a code refinement specialist designed to improve existing implementations based
on specific feedback. Analyze the provided feedback, identify areas for improvement,
and modify the code while strictly maintaining the required input / output formats and
component specifications .

=== Task Description ===
{ task_description }

=== Components ===
{components_description}

=== Overall Plan ===
{ plan_text }

=== Test Cases ===
{ sampled_test_cases }

=== User Feedback ===
{user_feedback}

=== Previous Best Code Generation ===
{sampled_codes_with_error_info}

=== Refinement Requirements ===
Before refining the code, tell me the reason why the last code failed to pass the test
 function , and how would you improve the code.

=== Instructions ===
Generate the COMPLETE code based on the components and plan above.
DO MAKE SURE the complete code is a runnable function, all components are correctly
 integrated with in this function .
The complete function should take the input arguments as specified in the overall plan
and return the output as specified .
Please add as much comments as possible to your code to explain the logic and any
 critical steps .

```
 Structure  your  response  as  follows :
<Think>
Explain  why does last  code  failed  to  pass  the  Test  Function , and how to  fix   it .
</Think>

Your code  here . DO make sure the  output  is  a  single  function  that   integrates   all
components.

<Planning>
A  detailed   step−by−step  explanation  of  the  code's  workflow.
</Planning>
<Main Function Name>
The name of the  main  function  that   integrates   all  components.
</Main Function  Name>
Provide  the  code  with  the  same  indicator  and  structure   as  shown in   Instructions . DO
NOT return any  test  cases  or  example usages  in  your  code!
```

Listing 19: LLM-CG: Prompt for code refinement.

## F  SAMPLE TM-GENERATED TEST CASES

This section provides concrete examples of test cases generated by the TM agent (SciCode example; first subproblem of problem 4, using Qwen3-32B), following the rebuttal supplementary material format.

```
<separator>

<Type>
 correctness
</Type>
<Planning>
This  test  case  validates  the  correctness  of  the  incomplete  Cholesky  factorization
 function  using  a  small symmetric  positive   definite   matrix  with known structure  and
 sparsity . The input  matrix  is :

$$
A = \begin{bmatrix}
4 & 0 & 2 \\
0 & 5 & 0 \\
2 & 0 & 6
\end{bmatrix}
$$

The expected  incomplete  Cholesky  factor  $K$ should be  sparse ,  preserving  the  zero
 pattern  from $A$. The exact  Cholesky  decomposition  would give :

$$
L = \begin{bmatrix}
2 & 0 & 0 \\
0 & \sqrt {5}  & 0 \\
1 & 0 & \sqrt {2}
\end{bmatrix}
$$

Since  $A_{13} = A_{31} = 2 \neq 0$,  the  corresponding   entries   in  $K$ are  non−zero.
```

We compute the first row:
− $L_{11} = \sqrt{4} = 2$
− Since $A_{21} = 0$, $L_{21} = 0$
− $L_{31} = (2 − 0)/2 = 1$

Second row:
− $L_{22} = \sqrt{5}$
− $A_{32} = 0$, so $L_{32} = 0$

Third row:
− Compute diagonal: $L_{33} = \sqrt{6 − L_{31}^2} = \sqrt{6 − 1} = \sqrt{5}$

So the expected $K$ is:

$$
K = \begin{bmatrix}
2 & 0 & 0 \\
0 & \sqrt{5} & 0 \\
1 & 0 & \sqrt{5}
\end{bmatrix}
$$

We will use this to validate the output.
</Planning>


```python
def test_case(func):
    import numpy as np

    # Define the input matrix A
    A = np.array([
        [4, 0, 2],
        [0, 5, 0],
        [2, 0, 6]
    ], dtype=float)

    # Expected K based on reasoning
    expected_K = np.array([
        [2, 0, 0],
        [0, np.sqrt(5), 0],
        [1, 0, np.sqrt(5)]
    ])

    # Call the function under test
    try:
        result_K = func(A)
    except Exception as e:
        return False, f"Test failed: exception raised during execution − {e}"

    # Check if result is a NumPy array
    if not isinstance(result_K, np.ndarray):
        return False, "Test failed: output is not a NumPy array"

    # Check shape matches input
    if result_K.shape != A.shape:
        return False, f"Test failed: output shape {result_K.shape} does not match input shape {A.shape}"

    # Check for lower triangular structure
    if not np.allclose(result_K[np.triu_indices_from(result_K, k=1)], 0):
```

```
        return False, "Test failed: upper triangle contains non−zero values"

    # Check sparsity pattern matches A's lower triangle
    if not np. allclose (result_K [np.where(A == 0)], 0):
        return False, "Test failed: non−zero entries in positions where input was zero"

    # Compare result with expected K using a tolerance due to floating point precision
    if not np. allclose (result_K, expected_K, atol =1e−6):
        return False, f"Test failed: expected {expected_K}, got {result_K}"

    return True, "Test passed: correct incomplete Cholesky factor computed"


<separator>

<Type>
edge_case
</Type>
<Planning>
This test case checks how the function handles an empty matrix input. Incomplete
Cholesky factorization is undefined for empty matrices. We expect the function to
either raise an error or return an appropriately sized empty matrix. This tests the
robustness of the implementation when given invalid or edge−case inputs.
</Planning>

def test_case (func):
    import numpy as np

    # Input is an empty matrix
    A = np. array ([], dtype= float ). reshape ((0, 0))

    try :
        result_K = func(A)
    except Exception as e:
        return False, f"Test failed: exception raised during execution − {e}"

    # Expecting an empty matrix of the same shape
    if result_K .shape != (0, 0):
        return False, f"Test failed: expected empty matrix but got shape
{result_K .shape}"

    return True, "Test passed: function correctly handled empty matrix input"


<separator>

<Type>
edge_case
</Type>
<Planning>
This test case evaluates the function's behavior when provided with a singular matrix
(not positive definite). Incomplete Cholesky factorization requires the matrix to be
symmetric and positive definite. A singular matrix (e.g., one with determinant zero)
will cause the square root of a negative number or division by zero in the algorithm.
The test expects the function to handle such cases gracefully − either by raising an
error or returning an appropriate message.
</Planning>

def test_case (func):
```

```
import numpy as np

# Singular matrix that is symmetric
A = np.array([
    [1, 2],
    [2, 4]
], dtype=float)

try:
    result_K = func(A)
except ValueError:
    return True, "Test passed: function correctly raised an error for singular
matrix"
except Exception as e:
    return False, f"Test failed: unexpected exception raised - {e}"

# If no exception is raised, check for NaNs or invalid outputs
if np.any(np.isnan(result_K)):
    return True, "Test passed: function returned NaNs for singular matrix"

return False, "Test failed: function did not detect singular matrix"

```

Listing 20: TM agent outputs with test case type, planning rationale, and executable test function.

