# OpenReview forum: "AI-for-Science Low-code Platform with Bayesian Adversarial Multi-Agent Framework"
_ICLR.cc/2026/Conference — ICLR 2026 Poster_

### Official Review · Reviewer_vXXH · 2025-10-28

**Soundness:** 3
**Presentation:** 3
**Contribution:** 3
**Rating:** 4
**Confidence:** 4

**Summary:**

This paper introduces a novel AI4S (AI for Science) Low-Code Platform powered by a Bayesian Adversarial Framework. The goal is to empower domain experts by translating high-level natural language prompts into executable, domain-specific requirements, eliminating the need for intricate prompt engineering. The framework contains three parts, including a Task Manager, a Solution Generator, and an Evaluator. The framework iteratively refines the prompt through Bayesian update rule. Results show that the framework works well when the user prompt is written by non-experts and it also works well on general code generation questions.

**Strengths:**

1. The paper is clearly written and the ideas are easy to follow
2. The code generation framework is novel and achieves SOTA performance on both AI4S benchmarks and code generation benchmarks
3. Extensive experiments have been conducted to show more in-depth details about the framework

**Weaknesses:**

1. For the SciCode benchmark, the baseline is somewhat unclear, and comparing it to only a single baseline seems insufficient.

**Questions:**

1. What would the token budget–performance trade-off look like compared to other methods?
2. Could you provide more insights into the other modules? For example, how important is it to design the TM prompts as shown in the appendix? Which pieces of information are most important?

---

> ### Author Response · Authors · 2025-11-23
>
> We thank the reviewer for the positive assessment of our paper’s clarity, novelty, and extensive experiments. We address your questions regarding baselines and analysis below.
>
> 1. **Baselines for SciCode (Weakness 1)**&#x20;
>
> Regarding the baselines for SciCode, we clarify that our baseline is the **Standard Zero-Shot** method, utilizing the exact same initial inputs as our framework. Given that SciCode is a recently released benchmark (Jul 2024), the primary objective of this experiment was to investigate the **universality and scalability** of our framework across varying backbone **model sizes**, rather than solely comparing against other agentic architectures. Table 1 demonstrates that our method provides significant and consistent performance boosts across all configurations. To ensure sufficient comparison against other SOTA agents, we conducted extensive experiments in **Table 2** (on ScienceAgentBench against OpenHands and Self-Debug) and **Appendix Table 3** (on HumanEval/MBPP against a wide range of code agents), which collectively validate our framework's superior effectiveness on both complex scientific and traditional code generation tasks.
>
> 2. **Token Budget and Performance Trade-off (Question 1)**
>
> - **Trade-off:** You are correct that our framework involves multiple iterations, leading to a higher token cost than a direct One-Shot generation. However, for scientific tasks, **correctness is paramount**.
>
> - **Analysis:** As shown in Figure 3, performance improves significantly with iterations. While the token cost increases linearly with iterations, the "Valid Execution Rate" (Table 2) jumps from \~40% to \~90%. In scientific discovery, the cost of a failed/hallucinated simulation code is far higher than the cost of extra input tokens. We will add a brief discussion on this "Cost vs. Reliability" trade-off and list the increased token budget as a limitation in the discussion part.
>
> 3. **Insights into Other Modules and TM Prompt Design (Question 2)**
>
> - **Insights into Other Modules:** Each module in our framework is crucial for overall performance, which is verified by multiple ablation experiments and case studies. For instance, the ablation study on Adversarial Test Cases (ATC) in Figure 3a shows that removing this module leads to a significant performance drop after the third iteration, as the system stops receiving dynamic challenges. Similarly, the Bayesian recursive co-updating mechanism is shown to be essential, with performance monotonically increasing across iterations as the code and test cases co-evolve (Figure 3b). Detailed specifications and logic for these modules are provided in Section 2 and Appendix B.
>
> - **Importance of TM Prompt Design:** The structured design of the Task Manager (TM) prompt is a key driver of robustness, effectively improving for vague user instructions. As illustrated in Figure 4, our framework maintains high performance even when user inputs lack expert detail ("Without Knowledge"), whereas baselines exhibit a large performance gap. This demonstrates that the TM's internal prompt structure is critical for self-refining basic requests into executable specifications (which enforces planning and sanity checks).
>
> - **Most Important Information:** Every section of the prompt contributes to the final output, but the most critical aspect is the extracted and refined domain constraints (e.g., physical laws or formulas) derived from the user's request. The key lies in the system's ability to extend a potentially vague initial input into a scientifically rigorous plan, as seen in the Beach Profile Prediction case where incorporating the Bruun and Dean models was essential for success (Appendix D1). Therefore, the explicit translation of high-level user intent into domain-specific logical requirements is the most impactful information component.
>
> We hope these responses address your concerns and justify the value of our contribution.

---

### Official Review · Reviewer_FjrG · 2025-11-02

**Soundness:** 3
**Presentation:** 1
**Contribution:** 3
**Rating:** 6
**Confidence:** 4

**Summary:**

The paper proposed a multi-agent framework for robust code generation. The robustness in the generated code is ensured by assigning roles to each agent in the framework, one agent sets increasingly difficult tests for the code while one aims to generate code to pass them. A third permits preemptive evaluation of generated code by predicting performance (by generalizing from existing data) rather than having to execute the code, saving time and computation resources.

**Strengths:**

- The underlying idea of gradually increasing the difficulty of tests that generated output must pass, where the update is done on the basis of actual external evaluation and not on the basis of potentially untrustworthy LLM assessment, is a very good one. And has been implemented well: being resource-efficient by evaluating only a subset of code generations.
- Strong empirical performance of the proposed framework, across a diverse range of testbeds and use-cases.

**Weaknesses:**

- It is very unclear *where* the “bayesian update rule” is used. As in Algorithm 1, it suggests that it is used to re-weight existing test cases to present increasingly challenging ones as iterations proceed—that is, for the final prompt curation as each iteration. As per Appendix B, the rule is used for selecting a candidate code for evaluation (hence, by the evaluator agent).
- Unclear algorithmic steps: Line 12 says the $\lambda$'s are the test case weights, however they never get used in subsequent steps. The text suggests that they are updated based on $S_3^t$, however, the distribution over **existing** prompts (used interchangeably with test cases, seemingly) gets updated.
- Missing details: In Appendix D.1, a case study of the proposed method vs Cursor and Windsurf is presented. But it is unclear what underlying base models are being used, which as per the results in Table 2, has a strong bearing on the final performance. Further, the framework will have *three* (same or different) base models while Cursor and Windsurf presumably work with just one. Which models are used by all these methods add important insight on the downstream performance. It is difficult to interpret the results and determine the effectiveness of the framework without these details.
- Appendix B talking about “code embeddings” and a general setup for how GPs work. Details about how the code is embedded are lacking? What model is used to encode the code into embeddings?

**Questions:**

Does one of the agents iteratively generate test cases over iterations? If so, how are the weights over existing test cases, denoted by $\lambda$ in Algorithm 1, used?

---

> ### Author Response · Authors · 2025-11-23
>
> We thank the reviewer for the constructive feedback and for appreciating the resource efficiency and strong empirical performance of our framework. We address the specific technical clarifications below.
>
> 1. **Location and Role of the Bayesian Update Rule (Weakness 1 & 2)**
>
> The framework employs Bayesian principles in two distinct but complementary steps.&#x20;
>
>   - **Prompt Distribution Update (Algorithm 1):** The core Bayesian update (Eq. 6) is used to update the probability distribution of Prompt Components (Test Cases and Sample Codes), whose weights are updated using a deterministic method as detailed in eq 2 and 3. This determines what we ask the LLM to generate next.
>
>   - **Performance Prediction (Appendix B):** To save tokens and time, we use a Gaussian Process (GP) as a surrogate model to predict the performance of generated candidates before full execution. This helps in selecting the best candidates to feed back into the Bayesian Prompt Update.
>
> 2. **Clarification on $\lambda$ (Weakness 2 & Question)**
>
>   - **$\lambda$ in Algorithm 1:** This was a notation error in the pseudo-code. $\lambda$ was intended to represent the normalized weights of the test cases (derived from their "difficulty" score $S\_1$, where $S\_1$ are initialize to 1 for every test cases at t=0).
>
>   - **Usage:** These weights are used in the **Adversarial Sampling** step—harder test cases (higher $S\_1$) have a higher probability ($\lambda$) of being selected for the next generation prompt. We will correct Algorithm 1 to explicitly show how $\lambda$ updates the prompt distribution.
>
>   - **Correcting Initialization (Line 12)** Instead of initializing weights $\lambda$, we initialize the difficulty scores to prevent confusion of notation.
>
>      **Original:** Initialize test case weights $\forall i, \lambda\_i \gets 1/M$
>
>      **Revised:** Initialize test case difficulty scores $\forall i, S\_1(i)^0 \gets 1$
>
>
> 3. **Base Models in Case Studies (Weakness 3)**
>
> To ensure fairness and interpretability, we standardized the base model across all comparisons. GPT-4o was used as the underlying engine for both the baselines (Cursor, Windsurf) and all agents within our proposed framework. This setup ensures that the observed performance differences result from our framework's design rather than model discrepancies. We will explicitly state this in the revised Appendix.
>
>
> 4. **Code Embeddings in Appendix B (Weakness 4)**
>
> We implemented code embedding using OpenAI's text-embedding-3-large model. We chose this model for its superior semantic retrieval capabilities and its ability to handle the structural complexity of codebases effectively compared to earlier embedding models. This detail will be added to Appendix B.
>
> We hope these clarifications regarding the algorithmic notation and experimental settings address your concerns, and we look forward to incorporating these corrections to enhance the technical precision of our final manuscript.

---

### Official Review · Reviewer_G9ac · 2025-11-03

**Soundness:** 3
**Presentation:** 2
**Contribution:** 3
**Rating:** 6
**Confidence:** 4

**Summary:**

The paper proposes a multi-agent framework for AI for science code generation, which is trainable via Bayesian adversarial learning. The multi-agent framework enables a logical breakdown with a task manager, solution generator and an evaluator, where each agent has dedicated capabilites. The key idea is that the task manager iteratively modifies the prompt and test cases to different difficulty levels conditioned on the previous attempt by the solution generator. This naturally fits the notion of adversarial learning where two different modules compete against each other to achieve an equilibrium.

**Strengths:**

1. Very neat idea that can be plugged into any existing agentic framework and LLMs.
2. By design, the framework allows for curriculum learning (although this is not explicitly mentioned by the authors) -- I think this is afFigure 5 strength where the difficulty level can be adjusted over iterations.

**Weaknesses:**

1. Line 276: "TM adapts its weights for future evaluation" -- what weights are being updated here. Make it clear that the Bayesian updates are over the choice of prompts/codes, and not over tokens. Also, sub-agents are not trained, so there are no weight updates to the LLMs.
2. Are the test cases part of the plan that goes through loop 1 in Fig 2? Ideally, the domain expert should also provide some feedback on test cases during the planning phase. It is possible that the LLM generating test cases only generate dummy tests especially if the task is really difficult.
3. Based on the example in D.2.3, the extra advice provided by the domain experts can be very specific modeling design ideas. How to ensure the extra advice do not divulge the ground-truth best solution?

**Questions:**

Address weaknesses above.

Additional comments:
1. What is S(j+) in equation 2?
2. Provide some outputs of TM agent with sample test cases.
1. Can you upload the entire code snippet in D.1.6?
2. You use different icons for plan and codes. Actually, the legend in Fig 2 maybe wrong -- the icon used to denote N parallel codes is marked as plan.
3. The distinction between loop 2 and loop 3 in Fig 2 should be explicitly discussed somewhere in Sec 2.

Minor nitpickings:
1. Typos in line 15 algorithm 1: "Equation equation 234", "Test sase"
2. Typo line 230: "in Loop 3 of fig:Diagram",

---

> ### Author Response · Authors · 2025-11-23
>
> We thank the reviewer for the insightful comments and for recognizing the soundness of our adversarial framework and its capability for curriculum learning. We address your concerns and questions below.
>
> 1. **Clarification on Bayesian Updates and "Weights" (Weakness 1)**
>
> We apologize for the confusion caused by the phrasing in Line 276. We explicitly clarify that we do not update the weights (parameters) of the LLM itself.
>
> Instead, the "weights" refer to the probability distribution used to sample the components of the prompt (specifically, the pairing of Test Cases and Sample Codes).
>
> * **Mechanism:** The framework maintains a distribution of $P(Prompt\_{i,j})$, where $i$ is a test case and $j$ is a sample code. The Bayesian update rule (Eq. 6) updates this distribution based on the posterior score $S\_3$.
>
> * **Goal:** This allows the Task Manager (TM) to "learn" which combinations of test cases and reference codes are most effective for the current problem state, effectively performing "Prompt Optimization" rather than model training. We will revise Line 276 and Section 2.5 to make this distinction crystal clear.
>
> 2. **Test Case Generation and Expert Feedback (Weakness 2)**
>
> * **Expert Role in Loop 1:** In the Planning phase (Loop 1), the domain expert (user) reviews the *Plan*. Since the initial test cases are derived from this Plan, the expert effectively provides a high-level constraint on them.
>
> * **Adversarial Evolution:** We agree that initial LLM-generated tests can potentially be "dummy tests" or trivial. This is exactly why we introduce the **Adversarial Loop (Loop 2)**. The TM acts as a "Challenger," iteratively assigning higher weights on "harder" test cases ($S\_1$ score, which evaluates how well the test case distinguishes between high-quality code and low-quality code) to break the current solution. This evolutionary process filters out dummy tests and converges toward robust edge cases, even if the initial tests were simple. We will clarify this dynamic in Section 2.2.
>
> 3. **Expert Advice and Ground Truth (Weakness 3)**
>
> In our framework, "Expert Advice" refers to domain knowledge (e.g., physical formulas, data constraints, or logic descriptions like "Use the Bruun model equation"), not the code implementation itself. These inputs correspond directly to the background fields provided in the SciCode benchmark dataset used in our experiments.
>
> * **Prevention of Leakage:** In the experiments (including D.2.3), the "advice" provided was strictly limited to domain context and formulas available in textbooks or documentation, which a non-coding scientist would know. It does not contain the solution code. In fact, to ensure fairness in the experiment, apart from the SciCode (With Knowledge) and additional case study experiments, we did not use expert knowledge at all. We will add a disclaimer in the Appendix about how we strictly filtered advice to ensure no ground truth code leakage occurred.
>
> 4. **Clarifications on Equations and Figures (Additional comments 1, 4, 5)**
>
> * **Eq. 2 $S(j^{\dagger})$:** This represents the score of the code $j^{\dagger}$ that failed the test case. The term is meant to calculate the discriminative power of the test case (the difference between the average score of codes that passed and those that failed).
>
> * **Figure 2 Legend:** Thank you for spotting this. We will correct the legend for Plan.
>
> * **Loop 2 vs. Loop 3:** We will add a dedicated paragraph in Section 2.1 distinguishing them:
>
>   * **Loop 2 (Test Case Update):** Focuses on the TM agent updating the "Hardness" of Test Cases (Adversarial).
>
>   * **Loop 3 (Code/Prompt Update):** Focuses on the SG agent and Bayesian update of the *Prompt distribution* to generate better solutions.
>
> 5. **Additional Material&#x73; (Additional comments 2, 3)**
>
> While some of this information was implicitly included in our supplementary materials, we acknowledge that these raw debug logs can be verbose and unstructured. We are happy to provide the specific details requested below and will incorporate them into the Appendix of the revised manuscript. (Due to the character limit for a single response, we will provide the requested information across multiple replies.)
>
> 6. **Typos&#x20;**
>
> We will correct the typos in Algorithm 1 (e.g., "Test sase", $\lambda$) and the citation format in the final revision.
>
> We hope these clarifications address your concerns, and we look forward to incorporating them to strengthen the final version of our paper.

---

> > ### Author Response · Authors · 2025-11-23
> >
> > ### **Outputs of TM agent with sample test cases (regrading to Additional comments 2)**
> >
> > This information can also been found in our code files(e.g. /AI4Stemp/log/20250417\_101751.log), but since these logs are used for bugging purposes, it is quite verbose. Here we provide an example of the outputs of TM agents for test cases on Scicode problems (first subproblem of problem4), using Qwen3-32B
> >
> > ```python
> > <separator>
> >
> > <Type>
> > correctness
> > </Type>
> > <Planning>
> > This test case validates the correctness of the incomplete Cholesky factorization function using a small symmetric positive definite matrix with known structure and sparsity. The input matrix is:
> >
> > $$
> > A = \begin{bmatrix}
> > 4 & 0 & 2 \\
> > 0 & 5 & 0 \\
> > 2 & 0 & 6
> > \end{bmatrix}
> > $$
> >
> > The expected incomplete Cholesky factor $K$ should be sparse, preserving the zero pattern from $A$. The exact Cholesky decomposition would give:
> >
> > $$
> > L = \begin{bmatrix}
> > 2 & 0 & 0 \\
> > 0 & \sqrt{5} & 0 \\
> > 1 & 0 & \sqrt{2}
> > \end{bmatrix}
> > $$
> >
> > Since $A_{13} = A_{31} = 2 \neq 0$, the corresponding entries in $K$ are non-zero.
> >
> > We compute the first row:
> > - $L_{11} = \sqrt{4} = 2$
> > - Since $A_{21} = 0$, $L_{21} = 0$
> > - $L_{31} = (2 - 0)/2 = 1$
> >
> > Second row:
> > - $L_{22} = \sqrt{5}$
> > - $A_{32} = 0$, so $L_{32} = 0$
> >
> > Third row:
> > - Compute diagonal: $L_{33} = \sqrt{6 - L_{31}^2} = \sqrt{6 - 1} = \sqrt{5}$
> >
> > So the expected $K$ is:
> >
> > $$
> > K = \begin{bmatrix}
> > 2 & 0 & 0 \\
> > 0 & \sqrt{5} & 0 \\
> > 1 & 0 & \sqrt{5}
> > \end{bmatrix}
> > $$
> >
> > We will use this to validate the output.
> > </Planning>
> > <Code>
> > def test_case(func):
> >     import numpy as np
> >
> >     # Define the input matrix A
> >     A = np.array([
> >         [4, 0, 2],
> >         [0, 5, 0],
> >         [2, 0, 6]
> >     ], dtype=float)
> >
> >     # Expected K based on reasoning
> >     expected_K = np.array([
> >         [2, 0, 0],
> >         [0, np.sqrt(5), 0],
> >         [1, 0, np.sqrt(5)]
> >     ])
> >
> >     # Call the function under test
> >     try:
> >         result_K = func(A)
> >     except Exception as e:
> >         return False, f"Test failed: exception raised during execution - {e}"
> >
> >     # Check if result is a NumPy array
> >     if not isinstance(result_K, np.ndarray):
> >         return False, "Test failed: output is not a NumPy array"
> >
> >     # Check shape matches input
> >     if result_K.shape != A.shape:
> >         return False, f"Test failed: output shape {result_K.shape} does not match input shape {A.shape}"
> >
> >     # Check for lower triangular structure
> >     if not np.allclose(result_K[np.triu_indices_from(result_K, k=1)], 0):
> >         return False, "Test failed: upper triangle contains non-zero values"
> >
> >     # Check sparsity pattern matches A's lower triangle
> >     if not np.allclose(result_K[np.where(A == 0)], 0):
> >         return False, "Test failed: non-zero entries in positions where input was zero"
> >
> >     # Compare result with expected K using a tolerance due to floating point precision
> >     if not np.allclose(result_K, expected_K, atol=1e-6):
> >         return False, f"Test failed: expected {expected_K}, got {result_K}"
> >
> >     return True, "Test passed: correct incomplete Cholesky factor computed"
> > </Code>
> >
> > <separator>
> >
> > <Type>
> > edge_case
> > </Type>
> > <Planning>
> > This test case checks how the function handles an empty matrix input. Incomplete Cholesky factorization is undefined for empty matrices. We expect the function to either raise an error or return an appropriately sized empty matrix. This tests the robustness of the implementation when given invalid or edge-case inputs.
> > </Planning>
> > <Code>
> > def test_case(func):
> >     import numpy as np
> >
> >     # Input is an empty matrix
> >     A = np.array([], dtype=float).reshape((0, 0))
> >
> >     try:
> >         result_K = func(A)
> >     except Exception as e:
> >         return False, f"Test failed: exception raised during execution - {e}"
> >
> >     # Expecting an empty matrix of the same shape
> >     if result_K.shape != (0, 0):
> >         return False, f"Test failed: expected empty matrix but got shape {result_K.shape}"
> >
> >     return True, "Test passed: function correctly handled empty matrix input"
> > </Code>
> >
> > <separator>
> > ```

---

> > > ### Author Response · Authors · 2025-11-23
> > >
> > > ## **Entire code snippet in D.1.6 (regrading to Additional comments 3)**
> > >
> > > In the original manuscript, we presented an excerpt to focus the reader's attention on the core logic—specifically how the framework addresses user needs and expert guidelines—without overwhelming the section with boilerplate code.
> > >
> > > Below is the extended code snippet for D.1.6. Due to the character limit, we present the core logic here by omitting standard imports and sections already detailed in the original D.1.6 (e.g., "Component 2"). We will include this full version in the Appendix and use color-coding (similar to the highlighting method used in Section D.1.3) to clearly distinguish the key components.
> > >
> > > ```python
> > > # Define the function to run the entire process
> > > def run_beach_profile_model(train_data_path, test_data_path):
> > >     # Component 1: Data Loader
> > >     def data_loader(train_path, test_path):
> > >         # Load datasets
> > >         train_df = pd.read_excel(train_path)
> > >         test_df = pd.read_excel(test_path)
> > >
> > >         # One-hot encode categorical features
> > >         encoder = OneHotEncoder(sparse=False)
> > >         direction_encoded_train = encoder.fit_transform(train_df[['Dominant Wave Direction']])
> > >         direction_encoded_test = encoder.transform(test_df[['Dominant Wave Direction']])
> > >
> > >         # Drop the original categorical column and add the encoded columns
> > >         train_df = train_df.drop(columns=['Dominant Wave Direction'])
> > >         test_df = test_df.drop(columns=['Dominant Wave Direction'])
> > >         train_df = train_df.join(pd.DataFrame(direction_encoded_train, columns=encoder.get_feature_names_out(['Dominant Wave Direction'])))
> > >         test_df = test_df.join(pd.DataFrame(direction_encoded_test, columns=encoder.get_feature_names_out(['Dominant Wave Direction'])))
> > >
> > >         # Normalize numerical features
> > >         numerical_cols = train_df.select_dtypes(include=['int64', 'float64']).columns
> > >         # Exclude variable 'y' and 'x'
> > >         numerical_cols = numerical_cols.drop(['y', 'x'])
> > >         scaler = StandardScaler()
> > >         train_df[numerical_cols] = scaler.fit_transform(train_df[numerical_cols])
> > >         test_df[numerical_cols] = scaler.transform(test_df[numerical_cols])
> > >
> > >         return train_df, test_df
> > >
> > >     # Component 2: Theoretical Model Integration
> > >
> > >     # Component 3: Deep Learning Model
> > >     def deep_learning_model(X_train, y_train):
> > >         # Define the model architecture
> > >         class SimpleNN(nn.Module):
> > >             def __init__(self, input_size):
> > >                 super(SimpleNN, self).__init__()
> > >                 self.layer1 = nn.Linear(input_size, 128)
> > >                 self.layer2 = nn.Linear(128, 64)
> > >                 self.layer3 = nn.Linear(64, 32)
> > >                 self.output = nn.Linear(32, 1)
> > >                 self.relu = nn.ReLU()
> > >
> > >             def forward(self, x):
> > >                 x = self.relu(self.layer1(x))
> > >                 x = self.relu(self.layer2(x))
> > >                 x = self.relu(self.layer3(x))
> > >                 x = self.output(x)
> > >                 return x
> > >
> > >         # Initialize and compile the model
> > >         model = SimpleNN(X_train.shape[1])
> > >         return model
> > >
> > >     # Component 4: Model Training and Evaluation
> > >     def model_training_evaluation(model, train_df, test_df):
> > >         # Split the data into features and target
> > >         X_train = train_df.drop(columns=['y']).to_numpy()
> > >         y_train = train_df['y'].to_numpy()
> > >
> > >         X_test = test_df.drop(columns=['y']).to_numpy()
> > >         y_test = test_df['y'].to_numpy()
> > >
> > >         # Convert arrays to PyTorch tensors
> > >         X_train_tensor = torch.tensor(X_train, dtype=torch.float32)
> > >         y_train_tensor = torch.tensor(y_train, dtype=torch.float32).view(-1, 1)
> > >
> > >         X_test_tensor = torch.tensor(X_test, dtype=torch.float32)
> > >         y_test_tensor = torch.tensor(y_test, dtype=torch.float32).view(-1, 1)
> > >
> > >         # Create dataloaders
> > >         train_dataset = TensorDataset(X_train_tensor, y_train_tensor)
> > >         train_loader = DataLoader(train_dataset, batch_size=32, shuffle=True)
> > >
> > >         # Define loss function and optimizer
> > >         criterion = nn.MSELoss()
> > >         optimizer = optim.Adam(model.parameters(), lr=0.001)
> > >
> > >         # Training loop
> > >         model.train()
> > >         for epoch in range(100):
> > >             for batch_X, batch_y in train_loader:
> > >                 optimizer.zero_grad()
> > >                 outputs = model(batch_X)
> > >                 loss = criterion(outputs, batch_y)
> > >                 loss.backward()
> > >                 optimizer.step()
> > >
> > >         # Evaluation metrics
> > >         model.eval()
> > >         with torch.no_grad():
> > >             predictions = model(X_test_tensor)
> > >             mse = criterion(predictions, y_test_tensor).item()
> > >             rmse = np.sqrt(mse)
> > >             mae = np.mean(np.abs(predictions.numpy() - y_test_tensor.numpy()))
> > >             r_squared = 1 - (np.sum(np.square(predictions.numpy() - y_test_tensor.numpy())) / np.sum(np.square(y_test_tensor.numpy() - np.mean(y_test_tensor.numpy()))))
> > >
> > >         return {'RMSE': rmse, 'MAE': mae, 'R2': r_squared}
> > > ```

---

> > > > ### Comment · Reviewer_G9ac · 2025-11-25
> > > >
> > > > Thanks for addressing my comments. I would keep my already high score.

---

### Author Response · Authors · 2025-12-03
**Global Response**

We sincerely thank the reviewers for their time and constructive feedback. We are encouraged by their recognition of our framework's novelty and the soundness of our approach. While we address specific technical questions in our individual responses, we summarize our core contributions and address global themes below.

### **Summary of Contribution**

Our Bayesian Adversarial Multi-agent Framework addresses the reliability gap in AI for Science (AI4S) code generation through two key innovations:

* **Adversarial Co-evolution with GP Mechanism:** Both LLM-generated tests and solutions are strengthened recursively using a deterministic process based on their last iteration's performance score. To mitigate evaluation costs, a **Gaussian Process (GP)** surrogate model acts as a filter, estimating performance based on structural similarity to guide the search space without executing every candidate.

* **Bayesian Prompt Updates:** We treat prompt engineering as a probabilistic optimization problem. The framework updates the distribution of prompt components to select the optimal combination of dynamically weighted Test Cases and generated Codes.

### **1.  Bayesian Algorithms**

Reviewers requested clarification on the Bayesian update rule and "weights." We clarify that our framework optimizes the generation process, not the internal parameters of the LLM.

* **Prompt Distribution Update (Algorithm 1):** Each test and code sample is scored based on its performance, and normalized scores (weights) are assigned as the likelihood for each sampled combination. For example, if a specific Test Case effectively distinguishes between high and low-quality code (high "hardness" score $S_1$), the Bayesian rule increases its selection probability. Therefore, the Bayesian algorithm allows us to sample an unseen combination if it is likely to generate better code.

* **Gaussian Process Surrogate (Appendix B):** To enhance efficiency, the GP acts as a surrogate during selection. Instead of executing every candidate (which is expensive), the GP estimates potential performance based on embedding similarity to previously tested codes. This prioritizes promising candidates for the intensive testing phase, balancing exploration with execution costs.

### **2. Experimental Design**

To address the inquiries regarding the rationale behind our baseline selections and the comprehensive validation of our framework, we structured our experimental design into distinct phases that function synergistically to demonstrate performance across varying levels of complexity and domain specificity:

* **General & Logic (Appendix C):** Results on HumanEval, MBPP, and APPS confirm our approach is competitive in general programming and superior in handling complex logic compared to recent code agents.

* **Scalability on Scientific task (Table 1 - SciCode):** The framework shows significant improvements across all model sizes. Notably, it remains robust even without expert knowledge inputs ("Without Knowledge" setting), delivering performance comparable to fully knowledge-guided settings.

* **Complex Workflows (Table 2 - ScienceAgentBench):** We achieve SOTA performance on challenging, multi-step scientific tasks, significantly outperforming other agents in Valid Execution Rate.

* **Real-World Case Studies (Appendix D):** Case studies on Beach Profile Prediction and Brain MRI Segmentation demonstrate how the framework refines and incorporates expert knowledge to enhance output quality in practical research.

### **3. Revisions to the Manuscript**

We commit to the following revisions in the final manuscript:

* **Algorithm 1:** We will correct the notation regarding $\lambda$ to clearly distinguish between test case difficulty scores ($S\_1$) and their normalized sampling weights.

* **Appendix Expansion:** We will add details on the code embedding model (OpenAI text-embedding-3-large) and include **generated test case examples** from the Task Manager to improve reproducibility.

* **Discussion:** We will add a discussion on the **Token Budget vs. Correctness** trade-off. We will highlight that while our iterative method consumes more tokens than zero-shot prompting, this cost is justified in scientific domains where simulation reliability is paramount.

We believe these clarifications and revisions robustly address the reviewers' concerns. We look forward to incorporating these suggestions to deliver a high-quality final manuscript.

---

### Meta-Review · Area_Chair_R5QR · 2026-01-06

**Summary:**

This paper presents a novel Bayesian adversarial multi-agent low-code platform for AI-for-Science (AI4S) code generation, addressing critical challenges of reliability, error propagation, and non-expert accessibility in scientific code development. The core innovations—adversarial co-evolution with Gaussian Process surrogate and Bayesian prompt optimization—are well-motivated and technically sound, with extensive experiments (spanning general programming benchmarks, scientific tasks, and real-world case studies) demonstrating SOTA performance and robustness.
Reviewers raised valid concerns regarding Bayesian update mechanics, notation clarity, experimental baselines, and module transparency. The authors’ rebuttal comprehensively addresses these issues: clarifying that "weights" refer to prompt component distribution probabilities (not LLM parameters), correcting notation errors, standardizing base models for fair comparisons, supplementing test case examples and code snippets, and expanding details on code embedding (OpenAI text-embedding-3-large) and experimental design. These revisions significantly enhance the paper’s rigor and reproducibility.
The work’s strengths—modularity (easily integrable with existing LLMs/agent frameworks), support for non-coding domain experts, and effective mitigation of scientific code’s high-stakes errors—align well with ICLR’s focus on impactful, innovative AI research. The token budget-performance trade-off is thoughtfully justified, reflecting practical considerations for scientific applications where correctness outweighs marginal token costs.
Given the authors’ successful resolution of reviewer concerns and the paper’s novel contributions to AI4S tooling, this work merits acceptance. Minor revisions (as outlined in the rebuttal) will further polish the manuscript, and we recommend its publication.

**Reviewer Concerns:**

None. All reviewer questions were comprehensively addressed with concrete revisions, clarifications, or additional information.

**Reviewer Scores:**

N/A

---

### Decision · Program_Chairs · 2026-01-26

Accept (Poster)